# Revisiting Deep Learning Models for Tabular Data

**Yury Gorishniy**[\*†‡]     **Ivan Rubachev**[†♣]     **Valentin Khrulkov**[†]     **Artem Babenko**[†♣]

† Yandex, Russia
‡ Moscow Institute of Physics and Technology, Russia
♣ National Research University Higher School of Economics, Russia

## Abstract

The existing literature on deep learning for tabular data proposes a wide range of novel architectures and reports competitive results on various datasets. However, the proposed models are usually not properly compared to each other and existing works often use different benchmarks and experiment protocols. As a result, it is unclear for both researchers and practitioners what models perform best. Additionally, the field still lacks effective baselines, that is, the easy-to-use models that provide competitive performance across different problems.

In this work, we perform an overview of the main families of DL architectures for tabular data and raise the bar of baselines in tabular DL by identifying two simple and powerful deep architectures. The first one is a ResNet-like architecture which turns out to be a strong baseline that is often missing in prior works. The second model is our simple adaptation of the Transformer architecture for tabular data, which outperforms other solutions on most tasks. Both models are compared to many existing architectures on a diverse set of tasks under the same training and tuning protocols. We also compare the best DL models with Gradient Boosted Decision Trees and conclude that there is still no universally superior solution. The source code is available at https://github.com/yandex-research/rtdl.

## 1   Introduction

Due to the tremendous success of deep learning on such data domains as images, audio and texts (Goodfellow et al., 2016), there has been a lot of research interest to extend this success to problems with data stored in tabular format. In these problems, data points are represented as vectors of heterogeneous features, which is typical for industrial applications and ML competitions, where neural networks have a strong non-deep competitor in the form of GBDT (Chen and Guestrin, 2016; Ke et al., 2017; Prokhorenkova et al., 2018). Along with potentially higher performance, using deep learning for tabular data is appealing as it would allow constructing multi-modal pipelines for problems, where only one part of the input is tabular, and other parts include images, audio and other DL-friendly data. Such pipelines can then be trained end-to-end by gradient optimization for all modalities. For these reasons, a large number of DL solutions were recently proposed, and new models continue to emerge (Arik and Pfister, 2020; Badirli et al., 2020; Hazimeh et al., 2020; Huang et al., 2020; Klambauer et al., 2017; Popov et al., 2020; Song et al., 2019; Wang et al., 2017, 2020).

Unfortunately, due to the lack of established benchmarks (such as ImageNet (Deng et al., 2009) for computer vision or GLUE (Wang et al., 2019a) for NLP), existing papers use different datasets for evaluation and proposed DL models are often not adequately compared to each other. Therefore, from the current literature, it is unclear what DL model generally performs better than others and whether GBDT is surpassed by DL models. Additionally, despite the large number of novel architectures, the field still lacks simple and reliable solutions that allow achieving competitive performance with moderate effort and provide stable performance across many tasks. In that regard, Multilayer

---

[*]Correspondence to: yura.gorishniy@phystech.edu

35th Conference on Neural Information Processing Systems (NeurIPS 2021).

Perceptron (MLP) remains the main simple baseline for the field, however, it does not always represent a significant challenge for other competitors.

The described problems impede the research process and make the observations from the papers not conclusive enough. Therefore, we believe it is timely to review the recent developments from the field and raise the bar of baselines in tabular DL. We start with a hypothesis that well-studied DL architecture blocks may be underexplored in the context of tabular data and may be used to design better baselines. Thus, we take inspiration from well-known battle-tested architectures from other fields and obtain two simple models for tabular data. The first one is a ResNet-like architecture (He et al., 2015) and the second one is FT-Transformer — our simple adaptation of the Transformer architecture (Vaswani et al., 2017) for tabular data. Then, we compare these models with many existing solutions on a diverse set of tasks under the same protocols of training and hyperparameters tuning. First, we reveal that none of the considered DL models can consistently outperform the ResNet-like model. Given its simplicity, it can serve as a strong baseline for future work. Second, FT-Transformer demonstrates the best performance on most tasks and becomes a new powerful solution for the field. Interestingly, FT-Transformer turns out to be a more universal architecture for tabular data: it performs well on a wider range of tasks than the more "conventional" ResNet and other DL models. Finally, we compare the best DL models to GBDT and conclude that there is still no universally superior solution.

We summarize the contributions of our paper as follows:

1. We thoroughly evaluate the main models for tabular DL on a diverse set of tasks to investigate their relative performance.

2. We demonstrate that a simple ResNet-like architecture is an effective baseline for tabular DL, which was overlooked by existing literature. Given its simplicity, we recommend this baseline for comparison in future tabular DL works.

3. We introduce FT-Transformer — a simple adaptation of the Transformer architecture for tabular data that becomes a new powerful solution for the field. We observe that it is a more universal architecture: it performs well on a wider range of tasks than other DL models.

4. We reveal that there is still no universally superior solution among GBDT and deep models.

## 2 Related work

**The "shallow" state-of-the-art** for problems with tabular data is currently ensembles of decision trees, such as GBDT (Gradient Boosting Decision Tree) (Friedman, 2001), which are typically the top-choice in various ML competitions. At the moment, there are several established GBDT libraries, such as XGBoost (Chen and Guestrin, 2016), LightGBM (Ke et al., 2017), CatBoost (Prokhorenkova et al., 2018), which are widely used by both ML researchers and practitioners. While these implementations vary in detail, on most of the tasks, their performances do not differ much (Prokhorenkova et al., 2018).

During several recent years, a large number of deep learning models for tabular data have been developed (Arik and Pfister, 2020; Badirli et al., 2020; Hazimeh et al., 2020; Huang et al., 2020; Klambauer et al., 2017; Popov et al., 2020; Song et al., 2019; Wang et al., 2017). Most of these models can be roughly categorized into three groups, which we briefly describe below.

**Differentiable trees.** The first group of models is motivated by the strong performance of decision tree ensembles for tabular data. Since decision trees are not differentiable and do not allow gradient optimization, they cannot be used as a component for pipelines trained in the end-to-end fashion. To address this issue, several works (Hazimeh et al., 2020; Kontschieder et al., 2015; Popov et al., 2020; Yang et al., 2018) propose to "smooth" decision functions in the internal tree nodes to make the overall tree function and tree routing differentiable. While the methods of this family can outperform GBDT on some tasks (Popov et al., 2020), in our experiments, they do not consistently outperform ResNet.

**Attention-based models.** Due to the ubiquitous success of attention-based architectures for different domains (Dosovitskiy et al., 2021; Vaswani et al., 2017), several authors propose to employ attention-like modules for tabular DL as well (Arik and Pfister, 2020; Huang et al., 2020; Song et al., 2019). In our experiments, we show that the properly tuned ResNet outperforms the existing attention-based

models. Nevertheless, we identify an effective way to apply the Transformer architecture (Vaswani et al., 2017) to tabular data: the resulting architecture outperforms ResNet on most of the tasks.

**Explicit modeling of multiplicative interactions.** In the literature on recommender systems and click-through-rate prediction, several works criticize MLP since it is unsuitable for modeling multiplicative interactions between features (Beutel et al., 2018; Qin et al., 2021; Wang et al., 2017). Inspired by this motivation, some works (Beutel et al., 2018; Wang et al., 2017, 2020) have proposed different ways to incorporate feature products into MLP. In our experiments, however, we do not find such methods to be superior to properly tuned baselines.

The literature also proposes some other architectural designs (Badirli et al., 2020; Klambauer et al., 2017) that cannot be explicitly assigned to any of the groups above. Overall, the community has developed a variety of models that are evaluated on different benchmarks and are rarely compared to each other. Our work aims to establish a fair comparison of them and identify the solutions that consistently provide high performance.

## 3 Models for tabular data problems

In this section, we describe the main deep architectures that we highlight in our work, as well as the existing solutions included in the comparison. Since we argue that the field needs strong easy-to-use baselines, we try to reuse well-established DL building blocks as much as possible when designing ResNet (section 3.2) and FT-Transformer (section 3.3). We hope this approach will result in conceptually familiar models that require less effort to achieve good performance. Additional discussion and technical details for all the models are provided in supplementary.

**Notation.** In this work, we consider supervised learning problems. $D=\{(x_i,\ y_i)\}_{i=1}^{n}$ denotes a dataset, where $x_i=(x_i^{(num)},\ x_i^{(cat)}) \in \mathbb{X}$ represents numerical $x_{ij}^{(num)}$ and categorical $x_{ij}^{(cat)}$ features of an object and $y_i \in \mathbb{Y}$ denotes the corresponding object label. The total number of features is denoted as $k$. The dataset is split into three disjoint subsets: $D = D_{train} \cup D_{val} \cup D_{test}$, where $D_{train}$ is used for training, $D_{val}$ is used for early stopping and hyperparameter tuning, and $D_{test}$ is used for the final evaluation. We consider three types of tasks: binary classification $\mathbb{Y} = \{0,\ 1\}$, multiclass classification $\mathbb{Y} = \{1,\ \ldots,\ C\}$ and regression $\mathbb{Y} = \mathbb{R}$.

### 3.1 MLP

We formalize the "MLP" architecture in Equation 1.

$$\begin{aligned} \mathtt{MLP}(x) &= \mathtt{Linear}\left(\mathtt{MLPBlock}\left(\ldots\left(\mathtt{MLPBlock}(x)\right)\right)\right) \\ \mathtt{MLPBlock}(x) &= \mathtt{Dropout}(\mathtt{ReLU}(\mathtt{Linear}(x))) \end{aligned} \tag{1}$$

### 3.2 ResNet

We are aware of one attempt to design a ResNet-like baseline (Klambauer et al., 2017) where the reported results were not competitive. However, given ResNet's success story in computer vision (He et al., 2015) and its recent achievements on NLP tasks (Sun and Iyyer, 2021), we give it a second try and construct a simple variation of ResNet as described in Equation 2. The main building block is simplified compared to the original architecture, and there is an almost clear path from the input to output which we find to be beneficial for the optimization. Overall, we expect this architecture to outperform MLP on tasks where deeper representations can be helpful.

$$\begin{aligned} \mathtt{ResNet}(x) &= \mathtt{Prediction}\left(\mathtt{ResNetBlock}\left(\ldots\left(\mathtt{ResNetBlock}\left(\mathtt{Linear}(x)\right)\right)\right)\right) \\ \mathtt{ResNetBlock}(x) &= x + \mathtt{Dropout}(\mathtt{Linear}(\mathtt{Dropout}(\mathtt{ReLU}(\mathtt{Linear}(\mathtt{BatchNorm}(x)))))) \\ \mathtt{Prediction}(x) &= \mathtt{Linear}\left(\mathtt{ReLU}\left(\mathtt{BatchNorm}(x)\right)\right) \end{aligned} \tag{2}$$

### 3.3 FT-Transformer

In this section, we introduce FT-Transformer (**F**eature **T**okenizer + **Transformer**) — a simple adaptation of the Transformer architecture (Vaswani et al., 2017) for the tabular domain. Figure 1 demonstrates the main parts of FT-Transformer. In a nutshell, our model transforms all features (categorical and numerical) to embeddings and applies a stack of Transformer layers to the embeddings. Thus, every Transformer layer operates on the *feature* level of *one* object. We compare FT-Transformer to conceptually similar AutoInt in section 5.2.

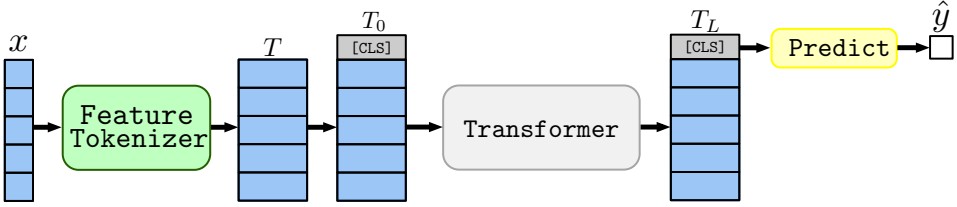

Figure 1: The FT-Transformer architecture. Firstly, Feature Tokenizer transforms features to embeddings. The embeddings are then processed by the Transformer module and the final representation of the [CLS] token is used for prediction.

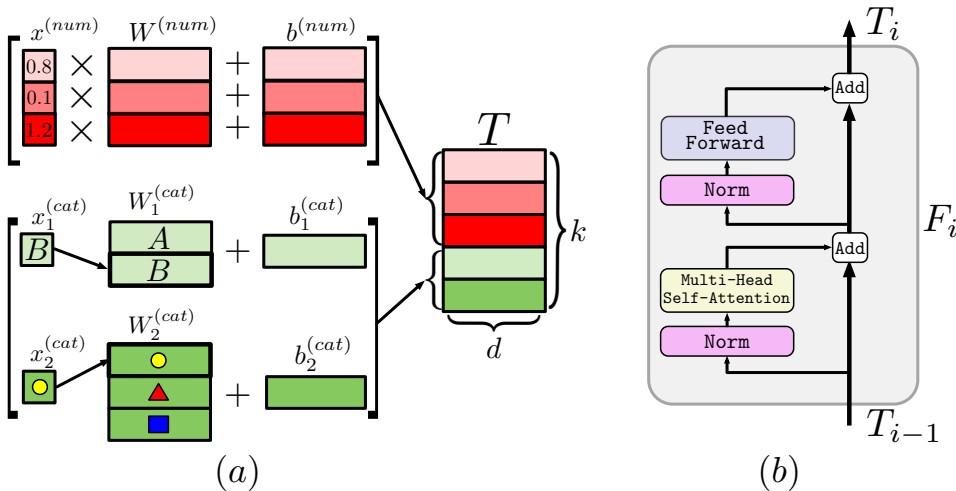

Figure 2: (a) Feature Tokenizer; in the example, there are three numerical and two categorical features; (b) One Transformer layer.

**Feature Tokenizer.** The Feature Tokenizer module (see Figure 2) transforms the input features $x$ to embeddings $T \in \mathbb{R}^{k \times d}$. The embedding for a given feature $x_j$ is computed as follows:

$$T_j = b_j + f_j(x_j) \in \mathbb{R}^d \qquad f_j : \mathbb{X}_j \to \mathbb{R}^d.$$

where $b_j$ is the $j$-th *feature bias*, $f_j^{(num)}$ is implemented as the element-wise multiplication with the vector $W_j^{(num)} \in \mathbb{R}^d$ and $f_j^{(cat)}$ is implemented as the lookup table $W_j^{(cat)} \in \mathbb{R}^{S_j \times d}$ for categorical features. Overall:

$$
\begin{aligned}
T_j^{(num)} &= b_j^{(num)} + x_j^{(num)} \cdot W_j^{(num)} && \in \mathbb{R}^d, \\
T_j^{(cat)} &= b_j^{(cat)} + e_j^T W_j^{(cat)} && \in \mathbb{R}^d, \\
T &= \mathtt{stack}\left[T_1^{(num)}, \ldots, T_{k^{(num)}}^{(num)}, T_1^{(cat)}, \ldots, T_{k^{(cat)}}^{(cat)}\right] && \in \mathbb{R}^{k \times d}.
\end{aligned}
$$

where $e_j^T$ is a one-hot vector for the corresponding categorical feature.

**Transformer.** At this stage, the embedding of the [CLS] token (or "classification token", or "output token", see Devlin et al. (2019)) is appended to $T$ and $L$ Transformer layers $F_1, \ldots, F_L$ are applied:

$$T_0 = \mathtt{stack}\left[[\mathtt{CLS}], T\right] \qquad T_i = F_i(T_{i-1}).$$

We use the PreNorm variant for easier optimization (Wang et al., 2019b), see Figure 2. In the PreNorm setting, we also found it to be necessary to remove the first normalization from the first Transformer layer to achieve good performance. See the original paper (Vaswani et al., 2017) for the background on Multi-Head Self-Attention (MHSA) and the Feed Forward module. See supplementary for details such as activations, placement of normalizations and dropout modules (Srivastava et al., 2014).

**Prediction.** The final representation of the [CLS] token is used for prediction:

$$\hat{y} = \texttt{Linear}(\texttt{ReLU}(\texttt{LayerNorm}(T_L^{\texttt{[CLS]}}))).$$

**Limitations.** FT-Transformer requires more resources (both hardware and time) for training than simple models such as ResNet and may not be easily scaled to datasets when the number of features is "too large" (it is determined by the available hardware and time budget). Consequently, widespread usage of FT-Transformer for solving tabular data problems can lead to greater CO2 emissions produced by ML pipelines, since tabular data problems are ubiquitous. The main cause of the described problem lies in the quadratic complexity of the vanilla MHSA with respect to the number of features. However, the issue can be alleviated by using efficient approximations of MHSA (Tay et al., 2020). Additionally, it is still possible to distill FT-Transformer into simpler architectures for better inference performance. We report training times and the used hardware in supplementary.

### 3.4 Other models

In this section, we list the existing models designed specifically for tabular data that we include in the comparison.

- **SNN** (Klambauer et al., 2017). An MLP-like architecture with the SELU activation that enables training deeper models.
- **NODE** (Popov et al., 2020). A differentiable ensemble of oblivious decision trees.
- **TabNet** (Arik and Pfister, 2020). A recurrent architecture that alternates dynamical reweighing of features and conventional feed-forward modules.
- **GrowNet** (Badirli et al., 2020). Gradient boosted weak MLPs. The official implementation supports only classification and regression problems.
- **DCN V2** (Wang et al., 2020). Consists of an MLP-like module and the feature crossing module (a combination of linear layers and multiplications).
- **AutoInt** (Song et al., 2019). Transforms features to embeddings and applies a series of attention-based transformations to the embeddings.
- **XGBoost** (Chen and Guestrin, 2016). One of the most popular GBDT implementations.
- **CatBoost** (Prokhorenkova et al., 2018). GBDT implementation that uses oblivious decision trees (Lou and Obukhov, 2017) as weak learners.

## 4 Experiments

In this section, we compare DL models to each other as well as to GBDT. Note that in the main text, we report only the key results. In supplementary, we provide: (1) the results for all models on all datasets; (2) information on hardware; (3) training times for ResNet and FT-Transformer.

### 4.1 Scope of the comparison

In our work, we focus on the relative performance of different architectures and do not employ various model-agnostic DL practices, such as pretraining, additional loss functions, data augmentation, distillation, learning rate warmup, learning rate decay and many others. While these practices can potentially improve the performance, our goal is to evaluate the impact of inductive biases imposed by the different model architectures.

### 4.2 Datasets

We use a diverse set of eleven public datasets (see supplementary for the detailed description). For each dataset, there is exactly one train-validation-test split, so all algorithms use the same splits. The datasets include: California Housing (CA, real estate data, Kelley Pace and Barry (1997)), Adult (AD, income estimation, Kohavi (1996)), Helena (HE, anonymized dataset, Guyon et al. (2019)),

Jannis (JA, anonymized dataset, Guyon et al. (2019)), Higgs (HI, simulated physical particles, Baldi et al. (2014); we use the version with 98K samples available at the OpenML repository (Vanschoren et al., 2014)), ALOI (AL, images, Geusebroek et al. (2005)), Epsilon (EP, simulated physics experiments), Year (YE, audio features, Bertin-Mahieux et al. (2011)), Covertype (CO, forest characteristics, Blackard and Dean. (2000)), Yahoo (YA, search queries, Chapelle and Chang (2011)), Microsoft (MI, search queries, Qin and Liu (2013)). We follow the pointwise approach to learning-to-rank and treat ranking problems (Microsoft, Yahoo) as regression problems. The dataset properties are summarized in Table 1.

Table 1: Dataset properties. Notation: "RMSE" ~ root-mean-square error, "Acc." ~ accuracy.

|  | CA | AD | HE | JA | HI | AL | EP | YE | CO | YA | MI |
|---|---|---|---|---|---|---|---|---|---|---|---|
| #objects | 20640 | 48842 | 65196 | 83733 | 98050 | 108000 | 500000 | 515345 | 581012 | 709877 | 1200192 |
| #num. features | 8 | 6 | 27 | 54 | 28 | 128 | 2000 | 90 | 54 | 699 | 136 |
| #cat. features | 0 | 8 | 0 | 0 | 0 | 0 | 0 | 0 | 0 | 0 | 0 |
| metric | RMSE | Acc. | Acc. | Acc. | Acc. | Acc. | Acc. | RMSE | Acc. | RMSE | RMSE |
| #classes | – | 2 | 100 | 4 | 2 | 1000 | 2 | – | 7 | – | – |

## 4.3 Implementation details

**Data preprocessing**. Data preprocessing is known to be vital for DL models. For each dataset, the same preprocessing was used for all deep models for a fair comparison. By default, we used the quantile transformation from the Scikit-learn library (Pedregosa et al., 2011). We apply standardization (mean subtraction and scaling) to Helena and ALOI. The latter one represents image data, and standardization is a common practice in computer vision. On the Epsilon dataset, we observed preprocessing to be detrimental to deep models' performance, so we use the raw features on this dataset. We apply standardization to regression targets for all algorithms.

**Tuning**. For every dataset, we carefully tune each model's hyperparameters. The best hyperparameters are the ones that perform best on the validation set, so the test set is never used for tuning. For most algorithms, we use the Optuna library (Akiba et al., 2019) to run Bayesian optimization (the Tree-Structured Parzen Estimator algorithm), which is reported to be superior to random search (Turner et al., 2021). For the rest, we iterate over predefined sets of configurations recommended by corresponding papers. We provide parameter spaces and grids in supplementary. We set the budget for Optuna-based tuning in terms of *iterations* and provide additional analysis on setting the budget in terms of *time* in supplementary.

**Evaluation**. For each tuned configuration, we run 15 experiments with different random seeds and report the performance on the test set. For some algorithms, we also report the performance of default configurations without hyperparameter tuning.

**Ensembles**. For each model, on each dataset, we obtain three ensembles by splitting the 15 single models into three disjoint groups of equal size and averaging predictions of single models within each group.

**Neural networks**. We minimize cross-entropy for classification problems and mean squared error for regression problems. For TabNet and GrowNet, we follow the original implementations and use the Adam optimizer (Kingma and Ba, 2017). For all other algorithms, we use the AdamW optimizer (Loshchilov and Hutter, 2019). We do not apply learning rate schedules. For each dataset, we use a predefined batch size for all algorithms unless special instructions on batch sizes are given in the corresponding papers (see supplementary). We continue training until there are `patience + 1` consecutive epochs without improvements on the validation set; we set `patience = 16` for all algorithms.

**Categorical features.** For XGBoost, we use one-hot encoding. For CatBoost, we employ the built-in support for categorical features. For Neural Networks, we use embeddings of the same dimensionality for all categorical features.

Table 2: Results for DL models. The metric values averaged over 15 random seeds are reported. See supplementary for standard deviations. For each dataset, top results are in **bold**. "Top" means "the gap between this result and the result with the best score is not statistically significant". For each dataset, ranks are calculated by sorting the reported scores; the "rank" column reports the average rank across all datasets. Notation: FT-T ~ FT-Transformer, ↓ ~ RMSE, ↑ ~ accuracy

| | CA ↓ | AD ↑ | HE ↑ | JA ↑ | HI ↑ | AL ↑ | EP ↑ | YE ↓ | CO ↑ | YA ↓ | MI ↓ | rank (std) |
|---|---|---|---|---|---|---|---|---|---|---|---|---|
| TabNet | 0.510 | 0.850 | 0.378 | 0.723 | 0.719 | 0.954 | 0.8896 | 8.909 | 0.957 | 0.823 | 0.751 | 7.5 (2.0) |
| SNN | 0.493 | 0.854 | 0.373 | 0.719 | 0.722 | 0.954 | 0.8975 | 8.895 | 0.961 | 0.761 | 0.751 | 6.4 (1.4) |
| AutoInt | 0.474 | **0.859** | 0.372 | 0.721 | 0.725 | 0.945 | 0.8949 | 8.882 | 0.934 | 0.768 | 0.750 | 5.7 (2.3) |
| GrowNet | 0.487 | **0.857** | – | – | 0.722 | – | 0.8970 | 8.827 | – | 0.765 | 0.751 | 5.7 (2.2) |
| MLP | 0.499 | 0.852 | 0.383 | 0.719 | 0.723 | 0.954 | 0.8977 | 8.853 | 0.962 | 0.757 | 0.747 | 4.8 (1.9) |
| DCN2 | 0.484 | 0.853 | 0.385 | 0.716 | 0.723 | 0.955 | 0.8977 | 8.890 | 0.965 | 0.757 | 0.749 | 4.7 (2.0) |
| NODE | 0.464 | **0.858** | 0.359 | 0.727 | 0.726 | 0.918 | 0.8958 | **8.784** | 0.958 | **0.753** | **0.745** | 3.9 (2.8) |
| ResNet | 0.486 | 0.854 | **0.396** | 0.728 | 0.727 | **0.963** | 0.8969 | 8.846 | 0.964 | 0.757 | 0.748 | 3.3 (1.8) |
| FT-T | **0.459** | **0.859** | 0.391 | **0.732** | **0.729** | 0.960 | **0.8982** | 8.855 | **0.970** | 0.756 | 0.746 | 1.8 (1.2) |

## 4.4 Comparing DL models

Table 2 reports the results for deep architectures.
**The main takeaways**:

- MLP is still a good sanity check
- ResNet turns out to be an effective baseline that none of the competitors can consistently outperform.
- FT-Transformer performs best on most tasks and becomes a new powerful solution for the field.
- Tuning makes simple models such as MLP and ResNet competitive, so we recommend tuning baselines when possible. Luckily, today, it is more approachable with libraries such as Optuna (Akiba et al., 2019).

Among other models, NODE (Popov et al., 2020) is the only one that demonstrates high performance on several tasks. However, it is still inferior to ResNet on six datasets (Helena, Jannis, Higgs, ALOI, Epsilon, Covertype), while being a more complex solution. Moreover, it is not a truly "single" model; in fact, it often contains significantly more parameters than ResNet and FT-Transformer and has an ensemble-like structure. We illustrate that by comparing *ensembles* in Table 3. The results indicate that FT-Transformer and ResNet benefit more from ensembling; in this regime, FT-Transformer outperforms NODE and the gap between ResNet and NODE is significantly reduced. Nevertheless, NODE remains a prominent solution among tree-based approaches.

Table 3: Results for ensembles of DL models with the highest ranks (see Table 2). For each model-dataset pair, the metric value averaged over three ensembles is reported. See supplementary for standard deviations. Depending on the dataset, the highest accuracy or the lowest RMSE is in **bold**. Due to the limited precision, some *different* values are represented with the same figures. Notation: ↓ ~ RMSE, ↑ ~ accuracy.

| | CA ↓ | AD ↑ | HE ↑ | JA ↑ | HI ↑ | AL ↑ | EP ↑ | YE ↓ | CO ↑ | YA ↓ | MI ↓ |
|---|---|---|---|---|---|---|---|---|---|---|---|
| NODE | 0.461 | 0.860 | 0.361 | 0.730 | 0.727 | 0.921 | 0.8970 | **8.716** | 0.965 | 0.750 | 0.744 |
| ResNet | 0.478 | 0.857 | 0.398 | 0.734 | 0.731 | 0.966 | 0.8976 | 8.770 | 0.967 | 0.751 | 0.745 |
| FT-Transformer | **0.448** | **0.860** | **0.398** | **0.739** | **0.731** | **0.967** | **0.8984** | 8.751 | **0.973** | **0.747** | **0.743** |

## 4.5 Comparing DL models and GBDT

In this section, our goal is to check whether DL models are *conceptually* ready to outperform GBDT. To this end, we compare the best possible metric values that one can achieve using GBDT or DL models, without taking speed and hardware requirements into account (undoubtedly, GBDT is a more lightweight solution). We accomplish that by comparing *ensembles* instead of single models since

GBDT is essentially an ensembling technique and we expect that deep architectures will benefit more from ensembling (Fort et al., 2020). We report the results in Table 4.

Table 4: Results for ensembles of GBDT and the main DL models. For each model-dataset pair, the metric value averaged over three ensembles is reported. See supplementary for standard deviations. Notation follows Table 3.

| | CA ↓ | AD ↑ | HE ↑ | JA ↑ | HI ↑ | AL ↑ | EP ↑ | YE ↓ | CO ↑ | YA ↓ | MI ↓ |
|---|---|---|---|---|---|---|---|---|---|---|---|
| | | | | | Default hyperparameters | | | | | | |
| XGBoost | 0.462 | **0.874** | 0.348 | 0.711 | 0.717 | 0.924 | 0.8799 | 9.192 | 0.964 | 0.761 | 0.751 |
| CatBoost | **0.428** | 0.873 | 0.386 | 0.724 | 0.728 | 0.948 | 0.8893 | 8.885 | 0.910 | 0.749 | 0.744 |
| FT-Transformer | 0.454 | 0.860 | **0.395** | **0.734** | **0.731** | **0.966** | **0.8969** | **8.727** | **0.973** | **0.747** | **0.742** |
| | | | | | Tuned hyperparameters | | | | | | |
| XGBoost | 0.431 | 0.872 | 0.377 | 0.724 | 0.728 | – | 0.8861 | 8.819 | 0.969 | **0.732** | 0.742 |
| CatBoost | **0.423** | **0.874** | 0.388 | 0.727 | 0.729 | – | 0.8898 | 8.837 | 0.968 | 0.740 | **0.741** |
| ResNet | 0.478 | 0.857 | 0.398 | 0.734 | 0.731 | 0.966 | 0.8976 | 8.770 | 0.967 | 0.751 | 0.745 |
| FT-Transformer | 0.448 | 0.860 | **0.398** | **0.739** | 0.731 | **0.967** | **0.8984** | **8.751** | **0.973** | 0.747 | 0.743 |

**Default hyperparameters**. We start with the default configurations to check the "out-of-the-box" performance, which is an important practical scenario. The default FT-Transformer implies a configuration with all hyperparameters set to some specific values that we provide in supplementary. Table 4 demonstrates that the ensemble of FT-Transformers mostly outperforms the ensembles of GBDT, which is not the case for only two datasets (California Housing, Adult). Interestingly, the ensemble of default FT-Transformers performs quite on par with the ensembles of tuned FT-Transformers.
**The main takeaway**: FT-Transformer allows building powerful ensembles out of the box.

**Tuned hyperparameters**. Once hyperparameters are properly tuned, GBDTs start dominating on some datasets (California Housing, Adult, Yahoo; see Table 4). In those cases, the gaps are significant enough to conclude that DL models do not universally outperform GBDT. Importantly, the fact that DL models outperform GBDT on most of the tasks does *not* mean that DL solutions are "better" in any sense. In fact, it only means that the constructed benchmark is slightly biased towards "DL-friendly" problems. Admittedly, GBDT remains an unsuitable solution to multiclass problems with a large number of classes. Depending on the number of classes, GBDT can demonstrate unsatisfactory performance (Helena) or even be untunable due to extremely slow training (ALOI).
**The main takeaways**:

- there is still no universal solution among DL models and GBDT
- DL research efforts aimed at surpassing GBDT should focus on datasets where GBDT outperforms state-of-the-art DL solutions. Note that including "DL-friendly" problems is still important to avoid degradation on such problems.

### 4.6 An intriguing property of FT-Transformer

Table 4 tells one more important story. Namely, FT-Transformer delivers most of its advantage over the "conventional" DL model in the form of ResNet exactly on those problems where GBDT is superior to ResNet (California Housing, Adult, Covertype, Yahoo, Microsoft) while performing on par with ResNet on the remaining problems. In other words, FT-Transformer provides competitive performance on all tasks, while GBDT and ResNet perform well only on some subsets of the tasks. This observation may be the evidence that FT-Transformer is a more "universal" model for tabular data problems. We develop this intuition further in section 5.1. Note that the described phenomenon is not related to ensembling and is observed for single models too (see supplementary).

## 5 Analysis

### 5.1 When FT-Transformer is better than ResNet?

In this section, we make the first step towards understanding the difference in behavior between FT-Transformer and ResNet, which was first observed in section 4.6. To achieve that, we design a

sequence of synthetic tasks where the difference in performance of the two models gradually changes from negligible to dramatic. Namely, we generate and *fix* objects $\{x_i\}_{i=1}^{n}$, perform the train-val-test split *once* and interpolate between two regression targets: $f_{GBDT}$, which is supposed to be easier for GBDT and $f_{DL}$, which is expected to be easier for ResNet. Formally, for one object:

$$x \sim \mathcal{N}(0, I_k), \qquad y = \alpha \cdot f_{GBDT}(x) + (1 - \alpha) \cdot f_{DL}(x).$$

where $f_{GBDT}(x)$ is an average prediction of 30 randomly constructed decision trees, and $f_{DL}(x)$ is an MLP with three randomly initialized hidden layers. Both $f_{GBDT}$ and $f_{DL}$ are generated once, i.e. the same functions are applied to all objects (see supplementary for details). The resulting targets are standardized before training. The results are visualized in Figure 3. ResNet and FT-Transformer perform similarly well on the ResNet-friendly tasks and outperform CatBoost on those tasks. However, the ResNet's relative performance drops significantly when the target becomes more GBDT friendly. By contrast, FT-Transformer yields competitive performance across the whole range of tasks.

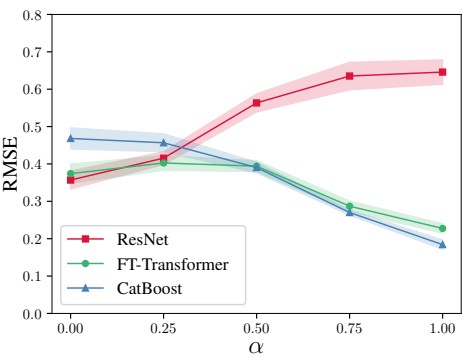

Figure 3: Test RMSE averaged over five seeds (shadows represent std. dev.). One $\alpha$ corresponds to one task; each task has the same set of train, validation and test features, but different targets.

The conducted experiment reveals a type of functions that are better approximated by FT-Transformer than by ResNet. Additionally, the fact that these functions are based on decision trees correlates with the observations in section 4.6 and the results in Table 4, where FT-Transformer shows the most convincing improvements over ResNet exactly on those datasets where GBDT outperforms ResNet.

## 5.2 Ablation study

In this section, we test some design choices of FT-Transformer.

First, we compare FT-Transformer with AutoInt (Song et al., 2019), since it is the closest competitor in its spirit. AutoInt also converts all features to embeddings and applies self-attention on top of them. However, in its details, AutoInt significantly differs from FT-Transformer: its embedding layer does not include feature biases, its backbone significantly differs from the vanilla Transformer (Vaswani et al., 2017), and the inference mechanism does not use the [CLS] token.

Second, we check whether feature biases in Feature Tokenizer are essential for good performance.

We tune and evaluate FT-Transformer without feature biases following the same protocol as in section 4.3 and reuse the remaining numbers from Table 2. The results averaged over 15 runs are reported in Table 5 and demonstrate both the superiority of the Transformer's backbone to that of AutoInt and the necessity of feature biases.

Table 5: The results of the comparison between FT-Transformer and two attention-based alternatives: AutoInt and FT-Transformer without feature biases. Notation follows Table 2.

| | CA $\downarrow$ | HE $\uparrow$ | JA $\uparrow$ | HI $\uparrow$ | AL $\uparrow$ | YE $\downarrow$ | CO $\uparrow$ | MI $\downarrow$ |
|---|---|---|---|---|---|---|---|---|
| AutoInt | 0.474 | 0.372 | 0.721 | 0.725 | 0.945 | 8.882 | 0.934 | 0.750 |
| FT-Transformer (w/o feature biases) | 0.470 | 0.381 | 0.724 | **0.727** | 0.958 | **8.843** | 0.964 | 0.751 |
| FT-Transformer | **0.459** | **0.391** | **0.732** | 0.729 | **0.960** | 8.855 | **0.970** | **0.746** |

### 5.3 Obtaining feature importances from attention maps

In this section, we evaluate attention maps as a source of information on feature importances for FT-Transformer for a given set of samples. For the $i$-th sample, we calculate the average attention map $p_i$ for the `[CLS]` token from Transformer's forward pass. Then, the obtained individual distributions are averaged into one distribution $p$ that represents the feature importances:

$$p = \frac{1}{n_{samples}} \sum_i p_i \qquad p_i = \frac{1}{n_{heads} \times L} \sum_{h,l} p_{ihl}.$$

where $p_{ihl}$ is the $h$-th head's attention map for the `[CLS]` token from the forward pass of the $l$-th layer on the $i$-th sample. The main advantage of the described heuristic technique is its efficiency: it requires a single forward for one sample.

In order to evaluate our approach, we compare it with Integrated Gradients (IG, Sundararajan et al. (2017)), a general technique applicable to any differentiable model. We use permutation test (PT, Breiman (2001)) as a reasonable interpretable method that allows us to establish a constructive metric, namely, rank correlation. We run all the methods on the train set and summarize results in Table 6. Interestingly, the proposed method yields reasonable feature importances and performs similarly to IG (note that this does not imply similarity to IG's feature importances). Given that IG can be orders of magnitude slower and the "baseline" in the form of PT requires $(n_{features} + 1)$ forward passes (versus one for the proposed method), we conclude that the simple averaging of attention maps can be a good choice in terms of cost-effectiveness.

Table 6: Rank correlation (takes values in $[-1, \ 1]$) between permutation test's feature importances ranking and two alternative rankings: Attention Maps (AM) and Integrated Gradients (IG). Means and standard deviations over five runs are reported.

|    | CA | HE | JA | HI | AL | YE | CO | MI |
|----|----|----|----|----|----|----|----|----|
| AM | 0.81 (0.05) | 0.77 (0.03) | 0.78 (0.05) | 0.91 (0.03) | 0.84 (0.01) | 0.92 (0.01) | 0.84 (0.04) | 0.86 (0.02) |
| IG | 0.84 (0.08) | 0.74 (0.03) | 0.75 (0.04) | 0.72 (0.03) | 0.89 (0.01) | 0.50 (0.03) | 0.90 (0.02) | 0.56 (0.02) |

## 6 Conclusion

In this work, we have investigated the status quo in the field of deep learning for tabular data and improved the state of baselines in tabular DL. First, we have demonstrated that a simple ResNet-like architecture can serve as an effective baseline. Second, we have proposed FT-Transformer — a simple adaptation of the Transformer architecture that outperforms other DL solutions on most of the tasks. We have also compared the new baselines with GBDT and demonstrated that GBDT still dominates on some tasks. The code and all the details of the study are open-sourced [1], and we hope that our evaluation and two simple models (ResNet and FT-Transformer) will serve as a basis for further developments on tabular DL.

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
