# Supplementary material

## A    Software and hardware

For most model-dataset pairs the workflow was as follows:

- tune the model on any suitable hardware
- evaluate the tuned model on one or more NVidia Tesla V100 32Gb

All the experiments were conducted under the same conditions in terms of software versions. For almost all experiments the used hardware can be found in the source code.

## B    Data

### B.1    Datasets

Table 1: Datasets description

| Name | Abbr | # Train | # Validation | # Test | # Num | # Cat | Task type | Batch size |
|---|---|---|---|---|---|---|---|---|
| California Housing | CA | 13209 | 3303 | 4128 | 8 | 0 | Regression | 256 |
| Adult | AD | 26048 | 6513 | 16281 | 6 | 8 | Binclass | 256 |
| Helena | HE | 41724 | 10432 | 13040 | 27 | 0 | Multiclass | 512 |
| Jannis | JA | 53588 | 13398 | 16747 | 54 | 0 | Multiclass | 512 |
| Higgs Small | HI | 62752 | 15688 | 19610 | 28 | 0 | Binclass | 512 |
| ALOI | AL | 69120 | 17280 | 21600 | 128 | 0 | Multiclass | 512 |
| Epsilon | EP | 320000 | 80000 | 100000 | 2000 | 0 | Binclass | 1024 |
| Year | YE | 370972 | 92743 | 51630 | 90 | 0 | Regression | 1024 |
| Covtype | CO | 371847 | 92962 | 116203 | 54 | 0 | Multiclass | 1024 |
| Yahoo | YA | 473134 | 71083 | 165660 | 699 | 0 | Regression | 1024 |
| Microsoft | MI | 723412 | 235259 | 241521 | 136 | 0 | Regression | 1024 |

### B.2    Preprocessing

For regression problems, we standardize the target values:

$$y_{new} = \frac{y_{old} - \texttt{mean}(y_{train}))}{\texttt{std}(y_{train})} \tag{1}$$

The feature preprocessing for DL models is described in the main text. Note that we add noise from $\mathcal{N}(0, 1e-3)$ to train numerical features for calculating the parameters (quantiles) of the quantile preprocessing as a workaround for features with few distinct values (see the source code for the exact implementation). The preprocessing is then applied to *original* features. We do not preprocess features for GBDTs, since this family of algorithms is insensitive to feature shifts and scaling.

## C    Results for all algorithms on all datasets

To measure statistical significance in the main text and in the tables in this section, we use the one-sided Wilcoxon (1945) test with $p = 0.01$.

Table 2 and Table 3 report all results for all models on all datasets.

Table 2: Results for single models with standard deviations. For each dataset, top results for baseline neural networks and FT-Transformer are in **blue**, the overall top results are in **red**. "Top" means "the gap between this result and the result with the best mean score is not statistically significant". "d" stands for "default". The remaining notation follows those from the main text.

| | CA↓ | AD↑ | HE↑ | JA↑ | HI↑ | AL↑ | EP↑ | YE↓ | CO↑ | YA↓ | MI↓ |
|---|---|---|---|---|---|---|---|---|---|---|---|
| | | | | | Baseline Neural Networks | | | | | | |
| TabNet | 0.510±7.6e-3 | 0.850±5.2e-3 | 0.378±1.7e-3 | 0.723±3.5e-3 | 0.719±1.7e-3 | 0.954±1.0e-3 | 0.8896±3.1e-3 | 8.909±2.3e-2 | 0.957±7.5e-3 | 0.823±9.2e-3 | 0.751±9.4e-4 |
| SNN | 0.493±4.6e-3 | 0.854±1.8e-3 | 0.373±2.8e-3 | 0.719±1.6e-3 | 0.722±2.2e-3 | 0.954±1.6e-3 | **0.8975**±**2.4e-4** | 8.895±1.9e-2 | 0.961±2.0e-3 | 0.761±5.3e-4 | 0.751±5.2e-4 |
| AutoInt | 0.474±3.3e-3 | **0.859**±**1.5e-3** | 0.372±2.5e-3 | 0.721±2.3e-3 | **0.725**±**1.7e-3** | 0.945±1.3e-3 | 0.8949±5.8e-4 | 8.882±3.3e-2 | 0.934±3.5e-3 | 0.768±1.1e-3 | 0.750±6.1e-4 |
| GrowNet | 0.487±7.1e-3 | **0.857**±**1.9e-3** | – | – | 0.722±1.6e-3 | – | 0.8970±5.7e-4 | 8.827±3.8e-2 | – | 0.765±1.2e-3 | 0.751±4.7e-4 |
| MLP | 0.499±2.9e-3 | 0.852±1.9e-3 | 0.383±2.6e-3 | 0.719±1.3e-3 | 0.723±1.8e-3 | 0.954±1.4e-3 | **0.8977**±**4.1e-4** | 8.853±3.1e-2 | 0.962±1.1e-3 | 0.757±3.5e-4 | 0.747±3.3e-4 |
| DCN2 | 0.484±2.4e-3 | 0.853±3.9e-3 | 0.385±3.0e-3 | 0.716±1.5e-3 | 0.723±1.3e-3 | 0.955±1.2e-3 | **0.8977**±**2.6e-4** | 8.890±2.8e-2 | **0.965**±**1.0e-3** | 0.757±1.9e-3 | 0.749±5.8e-4 |
| NODE | **0.464**±**1.5e-3** | **0.858**±**1.6e-3** | 0.359±2.0e-3 | 0.727±1.6e-3 | **0.726**±**1.3e-3** | 0.918±5.4e-3 | 0.8958±4.7e-4 | **8.784**±**1.6e-2** | 0.958±1.1e-3 | **0.753**±**2.5e-4** | **0.745**±**2.0e-4** |
| ResNet | 0.486±2.9e-3 | 0.854±1.7e-3 | **0.396**±**1.7e-3** | **0.728**±**1.5e-3** | **0.727**±**1.7e-3** | **0.963**±**7.5e-4** | 0.8969±4.4e-4 | 8.846±2.4e-2 | **0.964**±**1.1e-3** | 0.757±6.2e-4 | 0.748±3.1e-4 |
| | | | | | FT-Transformer | | | | | | |
| FT-Transformer_d | 0.469±3.8e-3 | 0.857±1.1e-3 | 0.381±2.4e-3 | 0.725±2.3e-3 | 0.725±1.8e-3 | 0.953±1.1e-3 | 0.8959±4.9e-4 | 8.889±4.6e-2 | 0.967±7.9e-4 | 0.756±8.2e-4 | 0.747±7.9e-4 |
| FT-Transformer | **0.459**±**3.5e-3** | **0.859**±**1.0e-3** | 0.391±1.2e-3 | **0.732**±**2.0e-3** | **0.729**±**1.5e-3** | 0.960±1.1e-3 | **0.8982**±**2.8e-4** | 8.855±3.1e-2 | **0.970**±**6.6e-4** | 0.756±8.2e-4 | 0.746±4.9e-4 |
| | | | | | GBDT | | | | | | |
| CatBoost_d | **0.430**±**7.4e-4** | 0.873±9.6e-4 | 0.381±1.5e-3 | 0.721±1.1e-3 | 0.726±8.0e-4 | 0.946±9.3e-4 | 0.8880±4.5e-4 | 8.913±5.5e-3 | 0.908±2.4e-4 | 0.751±2.0e-4 | 0.745±2.3e-4 |
| CatBoost | **0.431**±**1.5e-3** | 0.873±1.2e-3 | 0.385±1.1e-3 | 0.723±1.5e-3 | 0.725±1.5e-3 | – | 0.8880±5.8e-4 | 8.877±6.0e-3 | 0.966±2.7e-4 | 0.743±2.4e-4 | 0.743±2.1e-4 |
| XGBoost_d | 0.462±0.0 | **0.874**±**0.0** | 0.348±0.0 | 0.711±0.0 | 0.717±0.0 | 0.924±0.0 | 0.8799±0.0 | 9.192±0.0 | 0.964±0.0 | 0.761±0.0 | 0.751±0.0 |
| XGBoost | 0.433±1.6e-3 | 0.872±4.6e-4 | 0.375±1.2e-3 | 0.721±1.0e-3 | 0.727±1.0e-3 | – | 0.8837±1.2e-3 | 8.947±8.5e-3 | 0.969±5.1e-4 | **0.736**±**2.1e-4** | **0.742**±**1.3e-4** |

Table 3: Results for ensembles with standard deviations. Notation follows Table 2.

| | CA↓ | AD↑ | HE↑ | JA↑ | HI↑ | AL↑ | EP↑ | YE↓ | CO↑ | YA↓ | MI↓ |
|---|---|---|---|---|---|---|---|---|---|---|---|
| | | | | | Baseline Neural Networks | | | | | | |
| TabNet | 0.488±1.8e-3 | 0.856±3.4e-4 | 0.391±3.1e-4 | **0.736**±**1.3e-3** | 0.727±1.3e-3 | 0.961±2.8e-4 | 0.8944±6.8e-4 | 8.728±8.0e-3 | 0.966±1.5e-3 | 0.815±3.4e-3 | 0.746±3.5e-4 |
| SNN | 0.478±1.0e-3 | 0.857±3.1e-4 | 0.380±1.2e-3 | 0.727±8.7e-4 | 0.729±2.2e-3 | 0.962±2.8e-4 | 0.8976±7.5e-5 | 8.759±1.4e-3 | 0.966±4.5e-4 | 0.754±4.0e-4 | 0.747±5.2e-4 |
| AutoInt | **0.459**±**3.7e-3** | **0.860**±**2.2e-4** | 0.382±3.7e-4 | 0.733±7.8e-4 | **0.732**±**6.6e-4** | 0.959±1.7e-4 | 0.8966±2.5e-4 | 8.736±3.0e-3 | 0.950±1.1e-3 | 0.758±1.7e-4 | 0.747±1.5e-4 |
| GrowNet | 0.468±1.4e-3 | 0.859±6.3e-4 | – | – | 0.730±4.1e-4 | – | **0.8978**±**1.5e-4** | **8.683**±**6.6e-3** | – | 0.756±4.7e-4 | 0.747±1.4e-4 |
| MLP | 0.487±7.9e-4 | 0.855±4.8e-4 | 0.390±1.4e-3 | 0.725±2.1e-4 | 0.725±3.1e-4 | 0.960±3.2e-4 | **0.8979**±**1.1e-4** | 8.712±6.3e-3 | 0.966±9.1e-5 | 0.753±1.5e-4 | 0.746±1.4e-4 |
| DCN2 | 0.477±3.7e-4 | 0.857±3.2e-4 | 0.388±1.5e-4 | 0.719±1.5e-3 | 0.725±1.0e-3 | 0.960±4.1e-4 | 0.8977±4.8e-5 | 8.800±9.9e-3 | **0.969**±**6.4e-4** | 0.752±7.7e-4 | 0.746±1.4e-4 |
| NODE | 0.461±6.9e-4 | 0.860±4.0e-4 | 0.361±7.9e-4 | 0.730±8.4e-4 | 0.727±9.1e-4 | 0.921±1.6e-4 | 0.8970±3.7e-4 | 8.716±3.1e-3 | 0.965±5.0e-4 | **0.750**±**2.1e-5** | **0.744**±**8.2e-5** |
| ResNet | 0.478±7.9e-4 | 0.857±4.3e-4 | **0.398**±**7.2e-4** | 0.734±1.3e-3 | 0.731±8.5e-4 | **0.966**±**4.9e-4** | 0.8976±2.7e-4 | 8.770±8.0e-3 | 0.967±6.7e-4 | 0.751±7.5e-5 | 0.745±1.9e-4 |
| | | | | | FT-Transformer | | | | | | |
| FT-Transformer_d | 0.454±1.1e-3 | 0.860±4.9e-4 | 0.395±9.4e-4 | 0.734±7.5e-4 | 0.731±8.0e-4 | 0.966±3.9e-4 | 0.8969±1.9e-4 | 8.727±1.6e-2 | 0.973±3.2e-4 | **0.747**±**3.8e-4** | **0.742**±**3.3e-4** |
| FT-Transformer | **0.448**±**7.5e-4** | 0.860±3.9e-4 | **0.398**±**4.3e-4** | **0.739**±**5.9e-4** | 0.731±7.7e-4 | **0.967**±**4.8e-4** | **0.8984**±**1.6e-4** | 8.751±9.4e-3 | **0.973**±**1.1e-4** | 0.747±3.8e-4 | 0.743±1.1e-4 |
| | | | | | GBDT | | | | | | |
| CatBoost_d | 0.428±4.5e-5 | 0.873±4.2e-4 | 0.386±1.0e-3 | 0.724±4.8e-4 | 0.728±7.4e-4 | 0.948±9.2e-4 | 0.8893±2.7e-4 | 8.885±1.9e-3 | 0.910±3.0e-4 | 0.749±1.1e-4 | 0.744±4.4e-5 |
| CatBoost | **0.423**±**8.9e-4** | **0.874**±**4.5e-4** | 0.388±2.7e-4 | 0.727±6.4e-4 | 0.729±1.6e-3 | – | 0.8898±7.7e-5 | 8.837±3.2e-3 | 0.968±2.2e-5 | 0.740±1.7e-4 | **0.741**±**7.3e-5** |
| XGBoost_d | 0.462±0.0 | 0.874±0.0 | 0.348±0.0 | 0.711±0.0 | 0.717±0.0 | 0.924±0.0 | 0.8799±0.0 | 9.192±0.0 | 0.964±0.0 | 0.761±0.0 | 0.751±0.0 |
| XGBoost | 0.431±3.6e-4 | 0.872±2.3e-4 | 0.377±7.6e-4 | 0.724±3.4e-4 | 0.728±5.3e-4 | – | 0.8861±1.6e-4 | 8.819±4.0e-3 | 0.969±1.9e-4 | **0.732**±**5.4e-5** | 0.742±1.8e-5 |

# D Additional results

## D.1 Training times

Table 4: Training times in seconds averaged over 15 runs.

|              | CA   | AD   | HE   | JA   | HI   | AL   | EP   | YE   | CO   | YA    | MI   |
|--------------|------|------|------|------|------|------|------|------|------|-------|------|
| ResNet       | 72   | 144  | 363  | 163  | 91   | 933  | 704  | 777  | 4026 | 923   | 1243 |
| FT-Transformer | 187 | 128  | 536  | 576  | 257  | 2864 | 934  | 1776 | 5050 | 12712 | 2857 |
| Overhead     | 2.6x | 0.9x | 1.5x | 3.5x | 2.8x | 3.1x | 1.3x | 2.3x | 1.3x | 13.8x | 2.3x |

For most experiments, training times can be found in the source code. In Table 4, we provide the comparison between ResNet and FT-Transformer in order to "visualize" the overhead introduced by FT-Transformer compared to the main "conventional" DL baseline. The big difference on the Yahoo dataset is expected because of the large number of features (700).

## D.2 How tuning time budget affects performance?

In this section, we aim to answer the following questions:

- how does the relative performance of tuned models depends on tuning *time* budget?
- does the number of tuning iterations used in the main text allow models to reach most of their potential?

The first question is important for two main reasons. First, we have to make sure that longer tuning *times* of FT-Transformer (the number of tuning *iterations* is the same as for all other models) is not the reason of its strong performance. Second, we want to test FT-Transformer in the regime of low tuning time budget.

We consider four algorithms: XGBoost (as a fast GBDT implementation), MLP (as the fastest and simplest DL model), ResNet (as a stronger but slower DL model), FT-Transformer (as the strongest and the slowest DL model). We consider three datasets: California Housing, Adult, Higgs Small. On each dataset, for each algorithm, we run five independent (five random seeds) hyperparameter optimizations. Each run is constrained only by *time*. For each of the considered time budgets (15 minutes, 30 minutes, 1 hour, 2 hours, 3 hours, 4 hours, 5 hours, 6 hours), we pick the best model identified by Optuna on the validation set using no more than this time budget. Then, we report its performance and the number of Optuna iterations averaged over the five random seeds. The results are reported in Table 5. The takeaways are as follows:

- interestingly, FT-Transformer achieves good metrics just after several randomly sampled configurations (Optuna performs simple random sampling during the first 10 (default) iterations).
- FT-Transformer is slower to train, which is expected
- extended tuning (in terms of iterations) for other algorithms does not lead to any meaningful improvements

Table 5: Performance of tuned models with different tuning time budgets. Tuned model performance and the number of Optuna iterations (in parentheses) are reported (both metrics are averaged over five random seeds). Best results among DL models are in bold, overall best results are in bold red.

| | 0.25h | 0.5h | 1h | 2h | 3h | 4h | 5h | 6h |
|---|---|---|---|---|---|---|---|---|
| | | | | California Housing | | | | |
| XGBoost | **0.437 (31)** | **0.436 (56)** | **0.434 (120)** | **0.433 (252)** | **0.433 (410)** | **0.432 (557)** | **0.433 (719)** | **0.432 (867)** |
| MLP | 0.503(16) | 0.496(42) | 0.493(103) | 0.488(230) | 0.489(349) | 0.489(466) | 0.488(596) | 0.488(724) |
| ResNet | 0.488(7) | 0.487(15) | 0.483(30) | 0.481(64) | 0.482(101) | 0.482(131) | 0.482(164) | 0.484(197) |
| FT-Transformer | **0.466 (4)** | **0.464 (9)** | **0.465 (20)** | **0.460 (47)** | **0.458 (74)** | **0.458 (99)** | **0.457 (124)** | **0.459 (153)** |
| | | | | Adult | | | | |
| XGBoost | **0.871 (165)** | **0.873 (311)** | **0.872 (638)** | **0.872 (1296)** | **0.872 (1927)** | **0.872 (2478)** | **0.872 (2999)** | **0.872 (3500)** |
| MLP | 0.856(20) | 0.857(37) | 0.858(71) | 0.857(130) | 0.856(190) | 0.856(247) | 0.856(310) | 0.856(375) |
| ResNet | 0.856(8) | 0.854(16) | 0.854(32) | 0.856(69) | 0.855(105) | 0.855(140) | 0.856(174) | 0.855(208) |
| FT-Transformer | **0.861 (6)** | **0.860 (12)** | **0.859 (27)** | **0.859 (52)** | **0.860 (78)** | **0.860 (99)** | **0.860 (125)** | **0.860 (148)** |
| | | | | Higgs Small | | | | |
| XGBoost | 0.725(88) | 0.725(153) | 0.724(291) | 0.725(573) | 0.725(823) | 0.726(1069) | 0.725(1318) | 0.725(1559) |
| MLP | 0.721(16) | 0.720(29) | 0.723(62) | 0.722(137) | 0.724(220) | 0.723(300) | 0.724(375) | 0.724(447) |
| ResNet | 0.724(8) | 0.727(14) | 0.727(32) | 0.728(61) | 0.728(84) | 0.728(107) | 0.728(132) | 0.728(154) |
| FT-Transformer | **0.727 (2)** | **0.729 (5)** | **0.728 (12)** | **0.728 (23)** | **0.729 (34)** | **0.729 (44)** | **0.730 (56)** | **0.729 (66)** |

## E   FT-Transformer

In this section, we formally describe the details of FT-Transformer its tuning and evaluation. Also, we share additional technical experience and observations that were not used for final results in the paper but may be of interest to researchers and practitioners.

### E.1   Architecture

**Formal definition.**

$$\texttt{FT-Transformer}(x) = \texttt{Prediction}(\texttt{Block}(\ldots(\texttt{Block}(\texttt{AppendCLS}(\texttt{FeatureTokenizer}(x))))))$$

$$\texttt{Block}(x) = \texttt{ResidualPreNorm}(\texttt{FFN}, \texttt{ResidualPreNorm}(\texttt{MHSA}, x))$$
$$\texttt{ResidualPreNorm}(\texttt{Module}, x) = x + \texttt{Dropout}(\texttt{Module}(\texttt{Norm}(x)))$$
$$\texttt{FFN}(x) = \texttt{Linear}(\texttt{Dropout}(\texttt{Activation}(\texttt{Linear}(x))))$$

We use LayerNorm (Ba et al., 2016) as the normalization. See the main text for the description of Prediction and FeatureTokenizer. For MHSA, we set $n_{heads} = 8$ and do not tune this parameter.

**Activation.** Throughout the whole paper we used the ReGLU activation, since it is reported to be superior to the usually used GELU activation (Narang et al., 2021; Shazeer, 2020). However, we did not observe strong difference between ReGLU and ReLU in preliminary experiments.

**Dropout rates.** We observed that the attention dropout is always beneficial and FFN-dropout is also usually set by the tuning process to some non-zero value. As for the final dropout of each residual branch, it is rarely set to non-zero values by the tuning process.

**PreNorm vs PostNorm.** We use the PreNorm variant of Transformer, i.e. normalizations are placed at the beginning of each residual branch. The PreNorm variant is known for better optimization properties as opposed to the original Transformer, which is a PostNorm-Transformer (Liu et al., 2020; Nguyen and Salazar, 2019; Wang et al., 2019). The latter one may produce better models in terms of target metrics (Liu et al., 2020), but it usually requires additional modifications to the model and/or the training process, such as learning rate warmup or complex initialization schemes (Huang et al., 2020; Liu et al., 2020). While the PostNorm variant can be an option for practitioners seeking for the best possible model, we use the PreNorm variant in order to keep the optimization simple and same

for all models. Note that in the PostNorm formulation the `LayerNorm` in the "Prediction" equation (see the section "FT-Transformer" in the main text) should be omitted.

## E.2 The default configuration(s)

Table 6 describes the configuration of FT-Transformer referred to as "default" in the main text. Note that it includes hyperparameters for both the model and the optimization. In fact, the configuration is a result of an "educated guess" and we did not invest much resources in its tuning.

Table 6: Default FT-Transformer used in the main text.

| | | |
|---|---|---|
| Layer count | 3 | |
| Feature embedding size | 192 | |
| Head count | 8 | |
| Activation & FFN size factor | (ReGLU, $4/3$) | |
| Attention dropout | 0.2 | |
| FFN dropout | 0.1 | |
| Residual dropout | 0.0 | |
| Initialization | Kaiming | (He et al., 2015a) |
| Parameter count | 929K | The value is given for 100 numerical features |
| Optimizer | AdamW | |
| Learning rate | $1e{-}4$ | |
| Weight decay | $1e{-}5$ | 0.0 for Feature Tokenizer, LayerNorm and biases |

where "FFN size factor" is a ratio of the `FFN`'s hidden size to the feature embedding size.

We also designed a heuristic scaling rule to produce "default" configurations with the number of layers from one to six. We applied it on the Epsilon and Yahoo datasets in order to reduce the number of tuning iterations. However, we did not dig into the topic and our scaling rule may be suboptimal, see Wies et al. (2021) for a theoretically sound scaling rule.

In Table 7, we provide hyperparameter space used for Optuna-driven tuning (Akiba et al., 2019). For Epsilon, however, we iterated over several "default" configurations using a heuristic scaling rule, since the full tuning procedure turned out to be too time consuming. For Yahoo, we did not perform tuning at all, since the default configuration already performed well. In the main text, for FT-Transformer on Yahoo, we report the result of the default FT-Transformer.

Table 7: FT-Transformer hyperparameter space. Here (A) = {CA, AD, HE, JA, HI} and (B) = {AL, YE, CO, MI}

| Parameter | (Datasets) Distribution |
|---|---|
| # Layers | (A) UniformInt$[1, 4]$, (B) UniformInt$[1, 6]$ |
| Feature embedding size | (A,B) UniformInt$[64, 512]$ |
| Residual dropout | (A) $\{0, \text{Uniform}[0, 0.2]\}$, (B) Const$(0.0)$ |
| Attention dropout | (A,B) Uniform$[0, 0.5]$ |
| FFN dropout | (A,B) Uniform$[0, 0.5]$ |
| FFN factor | (A) Uniform$[2/3, 8/3]$, (B) Const$(4/3)$ |
| Learning rate | (A) LogUniform$[1e\text{-}5, 1e\text{-}3]$, (B) LogUniform$[3e\text{-}5, 3e\text{-}4]$ |
| Weight decay | (A,B) LogUniform$[1e\text{-}6, 1e\text{-}3]$ |
| # Iterations | (A) 100, (B) 50 |

## E.3 Training

On the Epsilon dataset, we scale FT-Transformer using the technique proposed by Wang et al. (2020) with the "headwise" sharing policy; we set the projection dimension to 128. We follow the popular

"transformers" library (Wolf et al., 2020) and do not apply weight decay to Feature Tokenizer, biases in linear layers and normalization layers.

## F  Models

In this section, we describe the implementation details for all models. See section E.1 for details on FT-Transformer.

### F.1  ResNet

**Architecture.** The architecture is formally described in the main text.

We tested several configurations and observed measurable difference in performance between all of them. We found the ones with "clear main path" (i.e. with all normalizations (except the last one) placed only in residual branches as in He et al. (2016) or Wang et al. (2019)) to perform better. As expected, it is also easier for them to train deeper configurations. We found the block design inspired by Transformer (Vaswani et al., 2017) to perform better or on par with the one inspired by the ResNet from computer vision (He et al., 2015b).

We observed that in the "optimal" configurations (the result of the hyperparameter optimization process) the inner dropout rate (not the last one) of one block was usually set to higher values compared to the outer dropout rate. Moreover, the latter one was set to zero in many cases.

**Implementation.** Ours, see the source code.

In Table 8, we provide hyperparameter space used for Optuna-driven tuning (Akiba et al., 2019).

Table 8: ResNet hyperparameter space. Here (A) = {CA, AD, HE, JA, HI, AL} and (B) = {EP, YE, CO, YA, MI}

| Parameter | (Datasets) Distribution |
|---|---|
| # Layers | (A) $\mathrm{UniformInt}[1, 8]$, (B) $\mathrm{UniformInt}[1, 16]$ |
| Layer size | (A) $\mathrm{UniformInt}[64, 512]$, (B) $\mathrm{UniformInt}[64, 1024]$ |
| Hidden factor | (A,B) $\mathrm{Uniform}[1, 4]$ |
| Hidden dropout | (A,B) $\mathrm{Uniform}[0, 0.5]$ |
| Residual dropout | (A,B) $\{0, \mathrm{Uniform}[0, 0.5]\}$ |
| Learning rate | (A,B) $\mathrm{LogUniform}[1e\text{-}5, 1e\text{-}2]$ |
| Weight decay | (A,B) $\{0, \mathrm{LogUniform}[1e\text{-}6, 1e\text{-}3]\}$ |
| Category embedding size | ({AD}) $\mathrm{UniformInt}[64, 512]$ |
| # Iterations | 100 |

### F.2  MLP

**Architecture.** The architecture is formally described in the main text.

**Implementation.** Ours, see the source code.

In Table 9, we provide hyperparameter space used for Optuna-driven tuning (Akiba et al., 2019). Note that the size of the first and the last layers are tuned and set separately, while the size for "in-between" layers is the same for all of them.

### F.3  XGBoost

**Implementation.** We fix and do not tune the following hyperparameters:

- `booster` $=$ `"gbtree"`
- `early-stopping-rounds` $= 50$
- `n-estimators` $= 2000$

Table 9: MLP hyperparameter space. Here (A) = {CA, AD, HE, JA, HI, AL} and (B) = {EP, YE, CO, YA, MI}

| Parameter | (Datasets) Distribution |
| --- | --- |
| # Layers | (A) UniformInt$[1, 8]$, (B) UniformInt$[1, 16]$ |
| Layer size | (A) UniformInt$[1, 512]$, (B) UniformInt$[1, 1024]$ |
| Dropout | (A,B) $\{0, \text{Uniform}[0, 0.5]\}$ |
| Learning rate | (A,B) LogUniform$[1e\text{-}5, 1e\text{-}2]$ |
| Weight decay | (A,B) $\{0, \text{LogUniform}[1e\text{-}6, 1e\text{-}3]\}$ |
| Category embedding size | ({AD}) UniformInt$[64, 512]$ |
| # Iterations | 100 |

In Table 10, we provide hyperparameter space used for Optuna-driven tuning (Akiba et al., 2019).

Table 10: XGBoost hyperparameter space. Here (A) = {CA, AD, HE, JA, HI} and (B) = {EP, YE, CO, YA, MI}

| Parameter | (Datasets) Distribution |
| --- | --- |
| Max depth | (A) UniformInt$[3, 10]$, (B) UniformInt$[6, 10]$ |
| Min child weight | (A,B) LogUniform$[1e\text{-}8, 1e5]$ |
| Subsample | (A,B) Uniform$[0.5, 1]$ |
| Learning rate | (A,B) LogUniform$[1e\text{-}5, 1]$ |
| Col sample by level | (A,B) Uniform$[0.5, 1]$ |
| Col sample by tree | (A,B) Uniform$[0.5, 1]$ |
| Gamma | (A,B) $\{0, \text{LogUniform}[1e\text{-}8, 1e2]\}$ |
| Lambda | (A,B) $\{0, \text{LogUniform}[1e\text{-}8, 1e2]\}$ |
| Alpha | (A,B) $\{0, \text{LogUniform}[1e\text{-}8, 1e2]\}$ |
| # Iterations | 100 |

## F.4 CatBoost

**Implementation.** We fix and do not tune the following hyperparameters:

- `early-stopping-rounds` $= 50$
- `od-pval` $= 0.001$
- `iterations` $= 2000$

In Table 11, we provide hyperparameter space used for Optuna-driven tuning (Akiba et al., 2019). We set the `task_type` parameter to "GPU" (the tuning was unacceptably slow on CPU).

**Evaluation.** We set the `task_type` parameter to "CPU", since for the used version of the CatBoost library it is crucial for performance in terms of target metrics.

## F.5 SNN

**Implementation.** Ours, see the source code.

In Table 12, we provide hyperparameter space used for Optuna-driven tuning (Akiba et al., 2019).

## F.6 NODE

**Implementation.** We used the official implementation: https://github.com/Qwicen/node.

Table 11: CatBoost hyperparameter space. Here (A) = {CA, AD, HE, JA, HI} and (B) = {EP, YE, CO, YA, MI}

| Parameter | (Datasets) Distribution |
|---|---|
| Max depth | (A) UniformInt$[3, 10]$, (B) UniformInt$[6, 10]$ |
| Learning rate | (A,B) LogUniform$[1e\text{-}5, 1]$ |
| Bagging temperature | (A,B) Uniform$[0, 1]$ |
| L2 leaf reg | (A,B) LogUniform$[1, 10]$ |
| Leaf estimation iterations | (A,B) UniformInt$[1, 10]$ |
| # Iterations | 100 |

Table 12: SNN hyperparameter space. Here (A) = {CA, AD, HE, JA, HI, AL} and (B) = {EP, YE, CO, YA, MI}

| Parameter | (Datasets) Distribution |
|---|---|
| # Layers | (A) UniformInt$[2, 16]$, (B) UniformInt$[2, 32]$ |
| Layer size | (A) UniformInt$[1, 512]$, (B) UniformInt$[1, 1024]$ |
| Dropout | (A,B) $\{0, \text{Uniform}[0, 0.1]\}$ |
| Learning rate | (A,B) LogUniform$[1e\text{-}5, 1e\text{-}2]$ |
| Weight decay | (A,B) $\{0, \text{LogUniform}[1e\text{-}5, 1e\text{-}3]\}$ |
| Category embedding size | ({AD}) UniformInt$[64, 512]$ |
| # Iterations | 100 |

**Tuning.** We iterated over the parameter grid from the original paper (Popov et al., 2020) plus the default configuration from the original paper. For multiclass datasets, we set the tree dimension being equal to the number of classes. For the Helena and ALOI datasets there was no tuning since NODE does not scale to classification problems with a large number of classes (for example, the minimal non-default configuration of NODE contains 600M+ parameters on the Helena dataset), so the reported results for these datasets are obtained with the default configuration.

### F.7  TabNet

**Implementation.** We used the official implementation:
https://github.com/google-research/google-research/tree/master/tabnet.
We always set `feature-dim` equal to `output-dim`. We also fix and do not tune the following hyperparameters (let A = {CA, AD}, B = {HE, JA, HI, AL}, C = {EP, YE, CO, YA, MI}):

- `virtual-batch-size` $= (A)\ 2048, (B)\ 8192, (C)\ 16384$
- `batch-size` $= (A)\ 256, (B)\ 512, (C)\ 1024$

In Table 13, we provide hyperparameter space used for Optuna-driven tuning (Akiba et al., 2019).

### F.8  GrowNet

**Implementation.** We used the official implementation: https://github.com/sbadirli/GrowNet. Note that it does not support multiclass problems, hence the gaps in the main tables for multiclass problems. We use no more than 40 small MLPs, each MLP has 2 hidden layers, boosting rate is learned – as suggested by the authors.

In Table 14, we provide hyperparameter space used for Optuna-driven tuning (Akiba et al., 2019).

Table 13: TabNet hyperparameter space.

| Parameter | Distribution |
|---|---|
| # Decision steps | $\mathrm{UniformInt}[3, 10]$ |
| Layer size | $\{8, 16, 32, 64, 128\}$ |
| Relaxation factor | $\mathrm{Uniform}[1, 2]$ |
| Sparsity loss weight | $\mathrm{LogUniform}[1e\text{-}6, 1e\text{-}1]$ |
| Decay rate | $\mathrm{Uniform}[0.4, 0.95]$ |
| Decay steps | $\{100, 500, 2000\}$ |
| Learning rate | $\mathrm{Uniform}[1e\text{-}3, 1e\text{-}2]$ |
| # Iterations | 100 |

Table 14: GrowNet hyperparameter space.

| Parameter | (Datasets) Distribution |
|---|---|
| Correct epochs | (all) $\{1, 2\}$ |
| Epochs per stage | (all) $\{1, 2\}$ |
| Hidden dimension | (all) $\mathrm{UniformInt}[32, 512]$ |
| Learning rate | (all) $\mathrm{LogUniform}[1e\text{-}5, 1e\text{-}2]$ |
| Weight decay | (all) $\{0, \mathrm{LogUniform}[1e\text{-}6, 1e\text{-}3]\}$ |
| Category embedding size | $(\{AD\})$ $\mathrm{UniformInt}[32, 512]$ |
| # Iterations | 100 |

## F.9 DCN V2

**Architecture.** There are two variats of DCN V2, namely, "stacked" and "parallel". We tuned and evaluated both and did not observe strong superiority of any of them. We report numbers for the "parallel" variant as it was slightly better on large datasets.

**Implementation.** Ours, see the source code.

In Table 15, we provide hyperparameter space used for Optuna-driven tuning (Akiba et al., 2019).

Table 15: DCN V2 hyperparameter space. Here (A) = {CA, AD, HE, JA, HI, AL} and (B) = {EP, YE, CO, YA, MI}

| Parameter | (Datasets) Distribution |
|---|---|
| # Cross layers | (A) $\mathrm{UniformInt}[1, 8]$, (B) $\mathrm{UniformInt}[1, 16]$ |
| # Hidden layers | (A) $\mathrm{UniformInt}[1, 8]$, (B) $\mathrm{UniformInt}[1, 16]$ |
| Layer size | (A) $\mathrm{UniformInt}[64, 512]$, (B) $\mathrm{UniformInt}[64, 1024]$ |
| Hidden dropout | (A,B) $\mathrm{Uniform}[0, 0.5]$ |
| Cross dropout | (A,B) $\{0, \mathrm{Uniform}[0, 0.5]\}$ |
| Learning rate | (A,B) $\mathrm{LogUniform}[1e\text{-}5, 1e\text{-}2]$ |
| Weight decay | (A,B) $\{0, \mathrm{LogUniform}[1e\text{-}6, 1e\text{-}3]\}$ |
| Category embedding size | $(\{AD\})$ $\mathrm{UniformInt}[64, 512]$ |
| # Iterations | 100 |

### F.10 AutoInt

**Implementation.** Ours, see the source code. We mostly follow the original paper (Song et al., 2019), however, it turns out to be necessary to introduce some modifications such as normalization in order to make the model competitive. We fix $n_{heads} = 2$ as recommended in the original paper.

In Table 16, we provide hyperparameter space used for Optuna-driven tuning (Akiba et al., 2019).

Table 16: AutoInt hyperparameter space. Here (A) = {CA, AD, HE, JA, HI} and (B) = {AL, YE, CO, MI}

| Parameter | (Datasets) Distribution |
|---|---|
| # Layers | (A,B) $\text{UniformInt}[1, 6]$ |
| Feature embedding size | (A,B) $\text{UniformInt}[8, 64]$ |
| Residual dropout | (A) $\{0, \text{Uniform}[0.0, 0.2]\}$, (B) $\text{Const}(0.0)$ |
| Attention dropout | (A,B) $\text{Uniform}[0.0, 0.5]$ |
| Learning rate | (A) $\text{LogUniform}[1e\text{-}5, 1e\text{-}3]$, (B) $\text{LogUniform}[3e\text{-}5, 3e\text{-}4]$ |
| Weight decay | (A,B) $\text{LogUniform}[1e\text{-}6, 1e\text{-}3]$ |
| # Iterations | (A) 100, (B) 50 |

## G Analysis

### G.1 When FT-Transformer is better than ResNet?

**Data.** Train, validation and test set sizes are $500\,000$, $50\,000$ and $100\,000$ respectively. One object is generated as $x \sim \mathcal{N}(0, I_{100})$. For each object, the first 50 features are used for target generation and the remaining 50 features play the role of "noise".

$f_{DL}$. The function is implemented as an MLP with three hidden layers, each of size 256. Weights are initialized with Kaiming initialization (He et al., 2015a), biases are initialized with the uniform distribution $\mathcal{U}(-a, \ a)$, where $a = d_{input}^{-0.5}$. All the parameters are fixed after initialization and are not trained.

$f_{GBDT}$. The function is implemented as an average prediction of 30 randomly constructed decision trees. The construction of one random decision tree is demonstrated in algorithm 1. The inference process for one decision tree is the same as for ordinary decision trees.

**CatBoost.** We use the default hyperparameters.

**FT-Transformer.** We use the default hyperparameters. Parameter count: $930K$.

**ResNet.** Residual block count: 4. Embedding size: 256. Dropout rate inside residual blocks: 0.5. Parameter count: $820K$.

### G.2 Ablation study

Table 17 is a more detailed version of the corresponding table from the main text.

Table 17: The results of the comparison between FT-Transformer and two attention-based alternatives. Means and standard deviations over 15 runs are reported

| | CA ↓ | HE ↑ | JA ↑ | HI ↑ | AL ↑ | YE ↓ | CO ↑ | MI ↓ |
|---|---|---|---|---|---|---|---|---|
| AutoInt | $0.474_{\pm 3.3e\text{-}3}$ | $0.372_{\pm 2.5e\text{-}3}$ | $0.721_{\pm 2.3e\text{-}3}$ | $0.725_{\pm 1.7e\text{-}3}$ | $0.945_{\pm 1.3e\text{-}3}$ | $8.882_{\pm 3.3e\text{-}2}$ | $0.934_{\pm 3.5e\text{-}3}$ | $0.750_{\pm 6.1e\text{-}4}$ |
| FT-Transformer (w/o feature biases) | $0.470_{\pm 5.7e\text{-}3}$ | $0.381_{\pm 1.6e\text{-}3}$ | $0.724_{\pm 3.9e\text{-}3}$ | $\mathbf{0.727}_{\pm 1.9e\text{-}3}$ | $0.958_{\pm 1.2e\text{-}3}$ | $\mathbf{8.843}_{\pm 2.5e\text{-}2}$ | $0.964_{\pm 6.2e\text{-}4}$ | $0.751_{\pm 5.6e\text{-}4}$ |
| FT-Transformer | $\mathbf{0.459}_{\pm 3.5e\text{-}3}$ | $\mathbf{0.391}_{\pm 1.2e\text{-}3}$ | $\mathbf{0.732}_{\pm 2.0e\text{-}3}$ | $\mathbf{0.729}_{\pm 1.5e\text{-}3}$ | $\mathbf{0.960}_{\pm 1.1e\text{-}3}$ | $8.855_{\pm 3.1e\text{-}2}$ | $\mathbf{0.970}_{\pm 6.6e\text{-}4}$ | $\mathbf{0.746}_{\pm 4.9e\text{-}4}$ |

**Algorithm 1:** Construction of one random decision tree.

**Result:** Random Decision Tree
set of leaves $L = \{\texttt{root}\}$;
`depths` - mapping from nodes to their depths;
`left` - mapping from nodes to their left children;
`right` - mapping from nodes to their right children;
`features` - mapping from nodes to splitting features;
`thresholds` - mapping from nodes to splitting thresholds;
`values` - mapping from leaves to their associated values;
$n = 0$ - number of nodes;
$k = 100$ - number of features;
**while** $n < 100$ **do**
    randomly choose leaf $z$ from $L$ s.t. $\texttt{depths}[z] < 10$;
    $\texttt{features}[z] \sim \texttt{UniformInt}[1, \ldots, k]$;
    $\texttt{thresholds}[z] \sim \mathcal{N}(0, 1)$;
    add two new nodes $l$ and $r$ to $L$;
    remove $z$ from $L$;
    unset $\texttt{values}[z]$;
    $\texttt{left}[z] = l$;
    $\texttt{right}[z] = r$;
    $\texttt{depths}[l] = \texttt{depths}[r] = \texttt{depths}[z] + 1$;
    $\texttt{values}[l] \sim \mathcal{N}(0, 1)$;
    $\texttt{values}[r] \sim \mathcal{N}(0, 1)$;
    $n = n + 2$;
**end**
return Random Decision Tree as $\{L, \texttt{left}, \texttt{right}, \texttt{features}, \texttt{thresholds}, \texttt{values}\}$.

# H   Additional datasets

Here, we report results for some datasets that turned out to be non-informative benchmarks, that is, where all models perform similarly. We report the average results over 15 random seeds for single models that are tuned and trained under the same protocol as described in the main text. The datasets include Bank (Moro et al., 2014), Kick [1], MiniBooNe [2], Click [3]. The dataset properties are given in Table 18 and the results are reported in Table 19.

Table 18: Additional datasets

| Dataset | # objects | # Num | # Cat | Task type (metric) |
|---|---|---|---|---|
| Bank | 45211 | 7 | 9 | Binclass (accuracy) |
| Kick | 72983 | 14 | 18 | Binclass (accuracy) |
| MiniBooNe | 130064 | 50 | 0 | Binclass (accuracy) |
| Click | 1000000 | 3 | 8 | Binclass (accuracy) |

---

[1] https://www.kaggle.com/c/DontGetKicked
[2] https://archive.ics.uci.edu/ml/datasets/MiniBooNE+particle+identification
[3] http://www.kdd.org/kdd-cup/view/kdd-cup-2012-track-2

Table 19: Results for single models on additional datasets.

| | Bank | Kick | MiniBooNE | Click |
|---|---|---|---|---|
| SNN | 0.9076 (0.0016) | 0.9014 (0.0007) | 0.9493 (0.0006) | 0.6613 (0.0006) |
| Grownet | 0.9093 (0.0012) | 0.9016 (0.0006) | 0.9494 (0.0007) | 0.6614 (0.0009) |
| DCNv2 | 0.9085 (0.0010) | 0.9014 (0.0007) | 0.9496 (0.0005) | 0.6615 (0.0003) |
| AutoInt | 0.9065 (0.0014) | 0.9005 (0.0005) | 0.9478 (0.0008) | 0.6614 (0.0005) |
| MLP | 0.9059 (0.0014) | 0.9012 (0.0004) | 0.9501 (0.0006) | 0.6617 (0.0006) |
| ResNet | 0.9072 (0.0014) | 0.9017 (0.0005) | 0.9508 (0.0006) | 0.6612 (0.0007) |
| FT-Transformer | 0.9090 (0.0014) | 0.9016 (0.0003) | 0.9491 (0.0007) | 0.6606 (0.0009) |
| FT-Transformer (default) | 0.9088 (0.0013) | 0.9013 (0.0006) | 0.9476 (0.0007) | 0.6610 (0.0007) |
| CatBoost | 0.9068 (0.0015) | 0.9021 (0.0009) | 0.9465 (0.0005) | 0.6635 (0.0002) |
| XgBoost | 0.9087 (0.0009) | 0.9034 (0.0003) | 0.9461 (0.0005) | 0.6399 (0.0006) |