# OpenReview forum: "Revisiting Deep Learning Models for Tabular Data"
_NeurIPS.cc/2021/Conference — NeurIPS 2021 Poster_

### Official Review · Reviewer_rxXz · 2021-07-12

**Rating:** 7
**Confidence:** 5

**Summary:**

The paper provides a comparison of recently proposed deep learning (DL) models for tabular data against established gradient-boosting decision tree (GBDT) methods.
The experimental evaluation shows that there is no clear winner between DL and GDBT models.

The paper also evaluates an MLP with residual connections ('ResNet') and introduces a Transformer-based model (FT-Transformer) similar to AutoInt.
The proposed architectures (ResNet/FT-Transformer) are demonstrated to give highly competitive performance, with FT-Transformer often performing SOTA.

**Ethical Concerns:**

None.

**Limitations And Societal Impact:**

The paper discusses limitations of the FT-Transformer in a paragraph in S. 3.6, where they identify the quadratic scaling in the number of features as the main limitation of the architecture.
Given that a main contribution of this paper is the comparison of architectures, a discussion of the limitations here would have been nice.

I believe the authors have misunderstood the request for discussing 'potential negative societal impacts' of submissions: in the checklist, they again reference S. 3.6 where they discuss only scaling limitations and how to (potentially) alleviate them in future work. It is unclear to me how this relates to societal impact.

**Main Review:**

## Strengths

I agree with the authors that there is a need for a thorough review and comparison of 'deep learning methods on tabular data'.
I like their overview of recent DL approaches in the introduction (and would have welcomed a more extensive overview).
The hyperparameter tuning of all involved models seems thorough at first impression.
The investigation of functional biases (S4.1) between GBDT and Neural Networks is an interesting idea.

## Weaknesses

Unfortunately, the weaknesses may outweigh the strengths for this submission.
The paper is uncomfortably split in two between (1) the comparison of established models and (2) the introduction of the FT-Transformer.
I believe that both parts would benefit from additional work.

### 1. Model Comparison

For a paper which lists the 'thorough' comparison of models on a 'wide range' of datasets as a key contribution,  the chosen selection of datasets is not adequate for a variety of reasons:
1. Only one of the datasets has categorical features. All other datasets have exclusively numerical features.
    * Categorical features are generally regarded as more challenging (especially for deep learning), so this omission may affect conclusions.
    * For this one dataset, the authors do not employ one hot encoding, which may negatively affect performance for some models.
2. 11 Datasets is definitely on the small side. (The SELU paper compares performance on 121 datasets.)
3. All datasets are between 20k and 1M samples.
    * There are a lot of UCI datasets with <20k samples. Why were they excluded? Surely these are relevant for such a general comparison?
    * Why were datasets cut to <1M samples? E.g. the Higgs dataset (usually 11M samples) was cut in a non-standard way to <1M samples.

Further,

4. one of the conclusions of the paper is that GBDT outperforms DL models on 'heterogeneous' but not on 'homogeneous' data.
    * The separation into homogeneous and heterogeneous datasets performed by the authors *perfectly* aligns with this conclusion.
    * However,  the specific reasons for how datasets where classified as homo-/heterogeneous is not explained in detail and feels rather arbitrary to me.
        * The authors state that a dataset is heterogenous if it describes 'different physical properties' and is of 'diverse units of measurements'. As far as I can tell, the Higgs dataset should be heterogenous by this definition, but I would be happy to hear explanations from the author. (Higgs contains a bunch of particle physics properties. Why are these less homogeneous then the housing-related attributes of California housing?)
        * For epsilon, I can't find any information online about the composition of its features, so why is it homogeneous then?
    * The perfect separation between GBDT and DL performance on heterogeneous vs homogeneous datasets as well as the lack of justification makes me suspicious that it is not impossible that *the authors came up with the classification of these datasets into hetero-/homogeneous only **after** observing the results*.
     * Also, no explanation of why performance of GDBT and DL differs on hetero-/homogeneous datasets is given.

### 2. Experiments

5.) For the Adult dataset, I have compared the reported performances in this paper to publically available results at https://www.openml.org/t/7592, which shows a large 0.1 difference between the top reported AUROC values. This might be indicative of problems with the experiment routine. I have not found such irregularities for a few other datasets that I've checked.

6.) The paper makes claims of superiority of one method (e.g ResNets / FT-Transformer) over another across all observed datasets.
Why not performance in terms of average rank order across models directly, instead of eyeballing things?
Additionally, I would have preferred for the authors to also quantify variance across different splits (instead just over different model seeds), especially for datasets without established test sets.

### 3. FT-Transformer

7.) The paper is split weirdly between 'model comparison/literature review' and introducing the 'FT-Transformer' architecture. Although both parts are individually clear, the introduction of the FT-Transformer architecture comes up somewhat abruptly in the middle of the paper, after the reader has already seen results for the FT-Transformer, affecting clarity of the overall submission.

I feel that making a big deal of the FT-Transformer as a contribution generally hurts this paper, because

* It takes away space that could have been used for a more thorough comparison of architectures
* The differences to AutoInt are small, despite what the authors claim. (Adding biases to the encoding sounds like a one line code change that seems to add a 0.01 improvement. Adding the mask tokens at input is more interesting but is not ablated.)

However, both the ResNet-MLP and the FT-Transformer architecture are potentially interesting as a baselines for future work and maybe the authors could have spent more time discussing the tricks needed to make them work well.
(Further, the conclusion that MLPs aren't horrible for tabular DL are in line with other recent work (https://arxiv.org/abs/2106.11189, previously at https://openreview.net/forum?id=2d34y5bRWxB).)

Overall, I feel like the split between 'review comparison of prior work' and 'introduction of new models'  does not work well in this submission.

### 4. Architecture Bias Experiments

The idea of randomly generating targets from GBDT/NN to study biases of the particular architectures is interesting.
However, I feel that the idea is underdeveloped in this submission:
 * What's benefit of linearly interpolating targets? (In my mind, this doesn't add anything as only $\alpha \in [0, 1]$ has a clear meaning.)
 * Why not generate data from the transformer as well?
 * Why is XGBoost not included here? Why not extend this to all models in the spirit of the 'big comparison' paper?
 * Fig. 3 seems to suggest that CatBoost can accommodate all data while ResNets can't acccomodate CatBoost-data. How does this relate to the experiments (where ostensibly, ResNet > CatBoost for some datasets).
 * Can this be related to the statements about hetero-/homogeneous datasets?

## Conclusion

Making a good literature review and model comparison paper is hard.

This paper is split between trying to provide that comparison and introducing the FT-Transformer.
I am not convinced that the FT-Transformer contains enough information to warrant publication.
The comparison between models might be enough if appropriately extended.
However, I feel that a significant number of the above comments need to be addressed first (mainly: more datasets, compare over different CV splits, show rank order, extend 4.1 to more methods).


[edit] Score updated to a 5. See discussion below.
[edit] Score updated to a 7. See discussion below.

**Time Spent Reviewing:**

5

---

> ### Author Response · Authors · 2021-08-10
> **Response to Reviewer rxxZ (Part 1)**
>
> Thank you for such a comprehensive review! Since there is a request for more datasets in several questions, we place the new results at the beginning of the response. The new datasets were picked as the ones from well-known repositories and used in the relevant literature. Note that this does not imply the same splits, so the metrics can be different from those from other papers.
>
> | Dataset   | #objects | #num-features | #cat-features | task type (metric)  |
> |:--------- |:-------- |:------------- |:------------- |:------------------- |
> | Bank      | 45211    | 7             | 9             | binclass (accuracy) |
> | Kick      | 72983    | 14            | 18            | binclass (accuracy) |
> | MiniBooNe | 130064   | 50            | 0             | binclass (accuracy) |
> | Click     | 1000000  | 3             | 8             | binclass (accuracy) |
>
> The tuning and evaluation protocols are the same as in the main text. The mean accuracy (with standard deviation in parentheses) over 15 random seeds is reported.
>
> |                          | Bank            |      Kick       |    MiniBooNE    |      Click      |
> |:------------------------ |:--------------- |:---------------:|:---------------:|:---------------:|
> | SNN                      | 0.9076 (0.0016) | 0.9014 (0.0007) | 0.9493 (0.0006) | 0.6613 (0.0006) |
> | Grownet                  | 0.9093 (0.0012) | 0.9016 (0.0006) | 0.9494 (0.0007) | 0.6614 (0.0009) |
> | DCNv2                    | 0.9085 (0.0010) | 0.9014 (0.0007) | 0.9496 (0.0005) | 0.6615 (0.0003) |
> | AutoInt                  | 0.9065 (0.0014) | 0.9005 (0.0005) | 0.9478 (0.0008) | 0.6614 (0.0005) |
> | MLP                      | 0.9059 (0.0014) | 0.9012 (0.0004) | 0.9501 (0.0006) | 0.6617 (0.0006) |
> | ResNet                   | 0.9072 (0.0014) | 0.9017 (0.0005) | 0.9508 (0.0006) | 0.6612 (0.0007) |
> | FT-Transformer           | 0.9090 (0.0014) | 0.9016 (0.0003) | 0.9491 (0.0007) | 0.6606 (0.0009) |
> | FT-Transformer (default) | 0.9088 (0.0013) | 0.9013 (0.0006) | 0.9476 (0.0007) | 0.6610 (0.0007) |
> | CatBoost                 | 0.9068 (0.0015) | 0.9021 (0.0009) | 0.9465 (0.0005) | 0.6635 (0.0002) |
> | XgBoost                  | 0.9087 (0.0009) | 0.9034 (0.0003) | 0.9461 (0.0005) | 0.6399 (0.0006) |
>
> Unfortunately, the considered datasets turned out to be non-informative benchmarks, since all algorithms perform similarly. Nevertheless, ResNet and FT-Transformer demonstrate competitive results, which confirms their potential to be effective baselines.
>
> ### 1. Model Comparison
>
> > Only one of the datasets has categorical features
>
> We have added three more datasets with categorical features, please, see the new results above.
>
> > For this one dataset, the authors do not employ one hot encoding, which may negatively affect performance for some models.
>
> We disagree. We used the best available strategy for each model:
> - XGBoost: One-Hot-Encoding
> - CatBoost: it has a built-in support for categorical features and we pass them as is
> - FT-Transformer: the section 3.6 in the main text describes. Essentially, One-Hot-Encoding is the core part of Feature Tokenizer in the context of categorical features
> - all other models, except MLP and NODE: embeddings, which also means One-Hot-Encoding under the hood
> - MLP, NODE: due to a bug, the strategy "replace categories with target-based counters" was used; we will fix the implementation, but there are no signs of possible changes in conclusions. Note that ResNet and other DL models use embeddings.
>
> > 11 Datasets is definitely on the small side. (The SELU paper compares performance on 121 datasets.)
>
> We agree that more datasets would result in more reliable conclusions. However, we also believe that the total number of datasets, after some reasonable amount, becomes less important than the *understanding of basic properties* of *each* of the selected datasets:
> 1. *Without* such understanding, it is hard to judge if those "121 datasets" are relevant to actual use cases and if those datasets are not biased in some unobvious way.
> 2. *With* such understanding, it becomes possible to make interesting observations, which led to POINT-4 in our case (please, see the global response above).
>
> > There are a lot of UCI datasets with <20k samples. Why were they excluded? Surely these are relevant for such a general comparison? Why were datasets cut to <1M samples?
>
> We feel that the paragraph "Datasets scope" (L154) is a bit misleading. Please, let us rephrase it, share our view and motivate our *intentional* choice of relatively high lower bound on the dataset size:
> 1. We indeed avoided small datasets (see the motivation below), but we did not "cut" datasets to 1M, it just happened that the largest dataset's size was 1M. In other words, we (1) believe that our conclusions do hold for datasets larger than 1M (2) admit it is possible that our conclusions do not hold for small datasets. So, in the "Datasets scope" paragraph we simply tried to make the reader aware of our choice.
> 2. We focused on the "bigger data" part of the spectrum because of the recent trends of increasingly bigger real-world datasets. For example, see the table below that compares shares of UCI datasets of different sizes between <2010 and >=2010 time periods. Note that we are aware of "small data" as well, but we believe that (1) in the "big data era" the scope of "bigger data" deserves more attention than before and (2) the >=10K~20K lower bound, as of today, can be considered as a reasonable choice in the context of Tabular Data (it is necessary to inform readers of this choice (and we aim to do that), but is not something that must be explicitly stated in the abstract, for example).
>
> Shares of UCI datasets of different sizes from different time periods:
>
> |         | 0-1K  | 1K-10K | 10K-100K | 100K-1M | 1M-inf | total |
> | :-----: | :---: | :----: | :------: | :-----: | :----: | :---: |
> | < 2010  | 0.53  |  0.28  |   0.14   |  0.05   |  0.03  |  145  |
> | >= 2010 | 0.37  |  0.28  |   0.21   |  0.09   |  0.07  |  391  |
>
> > the Higgs dataset (usually 11M samples) was cut in a non-standard way
>
> We disagree. We used the most popular distribution of this dataset in the OpenML repository (we are aware of at least one work "from the future" utilizing the same subset `[1]`).
>
> > However, the specific reasons for how datasets where classified as homo-/heterogeneous is not explained in detail and feels rather arbitrary to me
>
> We agree that the story about the "heterogeneous" data is not rigorous enough, thank you for pointing to this. It is indeed challenging to define the term "heterogeneous" and any attempt to provide such a definition will be too subjective. We plan to change this terminology to "DL/GBDT-friendly" and define the new terms in a utilitarian way (roughly speaking, "GBDT-friendly" means "GBDT outperforms conventional DL baselines (such as ResNet)", while "DL-friendly" means the opposite). Note that with this new terminology *all conclusions stay the same*.
>
> > Also, no explanation of why performance of GDBT and DL differs on hetero-/homogeneous datasets is given
>
> In this submission, we do not have such an explanation: it is a separate unsolved and, most importantly, **non-trivial** problem. It would be a huge step forward for the field if we gave such an explanation, but, as of now, our work provides only empirical evidence and highlights the benchmarks as well as a simple synthetic setup for future analysis.

---

> > ### Comment · Reviewer_rxXz · 2021-08-11
> > **Reviewer response to response to rebuttal**
> >
> > Dear authors,
> >
> > thanks for taking the time to write this extensive reply.
> >
> > ### 4. Architecture Bias Experiments
> >
> > > Yes, this is the main purpose of section 4.1 and we make this observation in L286-L288.
> >
> > There is a bit of a disconnect though, isn't there? It's unclear if (and in my opinion somewhat unlikely that) the GBDT-friendly vs. ResNet-friendly datasets really correspond to hetero vs. homogeneous datasets.
> >
> >
> > > It has a_direct_relation to societal impact. If tomorrow every ML pipeline working with Tabular Data is switched to FT-Transformer, we will see a tremendous relative increase in CO2 emissions generated by those pipelines.
> >
> > It would be great, if you could add that to the paper. I think the purpose of the societal-impact section is to explicitly point these things out – which, as far as I can tell, you are not doing in the current version.
> >
> > (Also, I'd be careful to claim that this will 'tremendously' increase the CO2 cost of these pipelines. I'm not sure how large the energy share of the model predictions is in these pipelines.)
> >
> > ### 3. FT-Transfomer
> >
> > > We feel that our message was misinterpreted. Please, see POINT-2, POINT-3 and POINT-4 in the global response above.
> >
> > In l. 62, you literally introduce the FT-Transformer as a major contribution of your paper. I feel like the phrasing of the paper that you present in the rebuttal is stronger than what you wrote in the paper. But I do feel that especially POINT-3 is not entirely consistent with the presentation in the paper.
> >
> > ### 2. Experiments
> >
> > >Thanks, we identify an issue in our pipeline that affected the results on Adult and we provide the updated results in the table below
> >
> > Glad I could help!
> >
> > > In fact, researchers come to similar conclusions for other fields as well (`[7]`, see section 3 and, in particular, section 3.2).
> >
> > Thanks for providing the reference. If I understand that reference correctly though, they argue *for* the inclusion of giving average rank  (and against averaging the metric directly)! See S. 3.2 that you link to:
> >
> > >  **To address this issue, one can use** geometric mean instead of arithmetic mean. There are also **solutions for rank aggregation** that ignore absolute score differences in favor of relative ordering [Dwork et al., 2001, Tabrizi et al., 2015\]: For instance, the **“average rank”** that is obtained by ranking the methods for each task based on their score and then computing the average ranks across tasks.
> >
> > I understand your points for including the results on each dataset (which I am also in favour of), however average rank order provides a quick way of judging which model performs 'best' on average.
> > If I want to choose a model for application to a new dataset, that is exactly the information I care about.
> >
> > Thanks for providing the ranks!
> >
> > Thanks for providing the experiments with the CV splits!
> >
> > > We feel that our message was misinterpreted. Please, see POINT-2, POINT-3 and POINT-4 in the global response above.
> >
> > It may be, that I have misinterpreted your message. Nevertheless, readers of the paper might feel similar to me, so I would appreciate it, if you could commit to making this less misleading in a future draft.
> >
> > > It is unclear to us, how this "0.01" number was obtained. Here are the differences inferred from Table 3 (re ~ regression, cl ~ classification):
> >
> > It seems to me as if the average across that table is roughly 0.01. Sorry, I should have been more specific.
> >
> > ### 1 Model Comparison
> >
> > Thanks for adding datasets with categorical input features!
> >
> > > For this one dataset, the authors do not employ one hot encoding, which may negatively affect performance for some models.
> > > > We disagree
> >
> > It would be great if you could clarify that in the text, because I remember trying to look up that information for a while.
> >
> > > We disagree. We used the most popular distribution of this dataset in the OpenML repository (we are aware of at least one work "from the future" utilizing the same subset`[1]`).
> >
> > I think you should at least clarify this.
> > Googling "Higgs Dataset" immediately leads to https://archive.ics.uci.edu/ml/datasets/HIGGS, which has 11 Mio rows.
> >
> > > We agree that the story about the "heterogeneous" data is not rigorous enough, thank you for pointing to this.
> >
> > Thanks for agreeing here!
> >
> > > In this submission, we do not have such an explanation
> >
> > That's entirely fine. I was just asking for your opinion.
> >
> > **Summary**
> > Thanks for your comprehensive answer and the new experiments confirmed. I feel that the experimental evaluation now meets the bar for publication.
> > The general idea of the paper is convincing and relevant (as evidenced by concurrent work).
> > However, I do feel that the presentation/writing of the paper does not present those ideas particularly clearly, and I am not convinced that the authors agree with me on this and are willing to fix this.
> > I do see that I am in the minority here, and will not strongly stand in the way of this getting published.
> > I think the discussion period will hopefully clear this up further!
> > Thanks!

---

> > > ### Author Response · Authors · 2021-08-13
> > > **Response to Reviewer rxXz**
> > >
> > > > There is a bit of a disconnect though, isn't there? It's unclear if (and in my opinion somewhat unlikely that) the GBDT-friendly vs. ResNet-friendly datasets really correspond to hetero vs. homogeneous datasets.
> > >
> > > In L286-L287, we write: *"... the fact that these functions are based on decision trees **correlates** with 287 the results in Table 1 and Table 2 ..."*. We intentionally use the term "correlation", so we do NOT claim that "synthetic GBDT-friendly data" and "real GBDT-friendly data" are the same things. Section 4.1 only demonstrates that the interesting empirical behavior of FT-Transformer can be observed in some other conditions, which (1) gives a hope that the phenomenon observed on real data is not a result of random coincidence (2) can serve as an inspiration for future work on understanding this behavior.
> > >
> > > > In l. 62, you literally introduce the FT-Transformer as a major contribution of your paper.
> > > >
> > > > ...
> > > >
> > > > It may be, that I have misinterpreted your message. Nevertheless, readers of the paper might feel similar to me, so I would appreciate it, if you could commit to making this less misleading in a future draft.
> > >
> > > FT-Transformer is neither the only nor the central contribution of our work. However, it is an important part of the overall picture, so we explicitly mention it in the summary. Nevertheless, it is still possible, that our wording is not perfect, we will be glad to hear suggestions in this regard!
> > >
> > > > It seems to me as if the average across that table is roughly 0.01. Sorry, I should have been more specific.
> > >
> > > We do not think that averaging differences across different tasks is a valid way of measuring performance.
> > >
> > > > It would be great if you could clarify that in the text, because I remember trying to look up that information for a while.
> > >
> > > We will add this information in the next revision.
> > >
> > > > I think you should at least clarify this. Googling "Higgs Dataset" immediately leads to https://archive.ics.uci.edu/ml/datasets/HIGGS, which has 11 Mio rows.
> > >
> > > We will do that in the next revision.
> > >
> > > > However, I do feel that the presentation/writing of the paper does not present those ideas particularly clearly, and I am not convinced that the authors agree with me on this and are willing to fix this.
> > >
> > > Not quite. We see that there are different opinions on the subject and we are actively thinking of possible improvements.

---

> > > > ### Comment · Reviewer_rxXz · 2021-08-25
> > > > **Response to authors**
> > > >
> > > > Thanks for your engagement.
> > > >
> > > > During the rebuttal, I feel that you have significantly changed major ideas/contributions of the paper.
> > > >
> > > > A) In the paper
> > > >
> > > > > We design a new attention-based architecture that outperforms the state-of-the-art DL models on tasks with heterogeneous data
> > > >
> > > > A) In the rebuttal:
> > > >
> > > > > we do NOT aim to propose conceptually novel architectures
> > > >
> > > > B) In the paper
> > > >
> > > > > Both Table 1 and Table 2 demonstrate that GBDT is strictly superior to DNN on heterogeneous data
> > > >
> > > > B) In the rebuttal
> > > >
> > > > > We agree that the story about the "heterogeneous" data is not rigorous enough, thank you for pointing to this. It is indeed challenging to define the term "heterogeneous" and any attempt to provide such a definition will be too subjective. We plan to change this terminology to "DL/GBDT-friendly" and define the new terms in a utilitarian way
> > > >
> > > > I believe both changes (A-B) are correct, but following up on them would correspond to a significant rewrite of the paper.
> > > >
> > > > Given the updated experiments, I have raised my score to a 5.
> > > >
> > > > _If you could provide me with drafts of all passages that you intend to rewrite to accommodate changes (A-B), I would be willing to further raise my score._
> > > >
> > > > Thank you!

---

> > > > > ### Author Response · Authors · 2021-08-26
> > > > > **Response to Reviewer rxXz**
> > > > >
> > > > > Thank you for the comment!
> > > > >
> > > > > Below, we address points (A) and (B). We propose *minor* changes to the text (on the level of phrases) that do not affect contributions and, hopefully, make our messages more accurate and clear.
> > > > >
> > > > > ### (A)
> > > > > There are several places where we use phrases like "we design a new attention-based architecture" (examples: L15, L50, L62). We agree that such words may cause too big expectations (and we did not want to cause such expectations), so our plan is to say "we design a (simple) adaptation of the Transformer architecture". The new variant seems to precisely describe the reality. What do you think?
> > > > >
> > > > > ### (B)
> > > > > In the paper, we use the informal terms "heterogeneous"/"homogeneous" in the hope of giving additional intuition on the difference between GBDT-friendly and DL-friendly datasets. In the discussion above, we came to the conclusion that informality is the issue. The good news is that this terminology itself is not a contribution and is not used to build any other ideas on top of it. It means that we can stop using this terminology without affecting contributions and messages. The specific changes include:
> > > > > - L22
> > > > >     - Before: "In these problems, data points are represented as vectors of heterogeneous features, which is typical for..."
> > > > >     - After: "Such problems are typical for...".
> > > > >     - Comment: this change is probably optional since in this case, we use the word "heterogeneous" in its "vocabulary" sense without any "mathematical" meaning.
> > > > > - L45
> > > > >     - Before: "For instance, GBDT is superior for data with heterogeneous features, but lags behind DL solutions on non-heterogeneous data and poorly scales to classification problems when the number of classes is large (1K+)"
> > > > >     - After: "" (this sentence can be removed)
> > > > > - L50
> > > > >     - Before: "Based on these observations, we design a new attention-based model that outperforms the ResNet baseline on tasks with heterogeneous features, providing a new state-of-the-art DL solution for most of the problems"
> > > > >     - After: "Additionally, we design a simple adaptation of the Transformer architecture for tabular data that turns out to be especially effective on the GBDT-friendly problems"
> > > > > - L62
> > > > >     - Before: "We design a new attention-based architecture that outperforms the state-of-the-art DL models on tasks with heterogeneous data.
> > > > >     - After: "We design an adaptation of the Transformer architecture for tabular data that demonstrates state-of-the-art performance among DL models on the problems where GBDT currently dominates."
> > > > > - L142-L151
> > > > >     - Before: "We informally call the dataset’s features “heterogeneous” ... "
> > > > >     - After: (option 1) remove all those sentences (option 2) keep the description of dataset domains without categorizing them
> > > > > - Tables 1-2
> > > > >     - Before: "Datasets with heterogeneous features are underlined"
> > > > >     - After: "" (the underlining can be removed)
> > > > > - L200
> > > > >     - Before: "Both Table 1 and Table 2 demonstrate that GBDT is strictly superior to DNN on heterogeneous data (see section 3.3), while on non-heterogeneous data situation is rather the opposite."
> > > > >     - After: "Both Table 1 and Table 2 demonstrate that the relative performance of DL models and GBDT strongly depends on a task."
> > > > > - L207-L208
> > > > >     - Before: "DL research efforts aimed at surpassing GBDT should cover a diverse set of heterogeneous datasets in addition to non-heterogeneous datasets, which should serve only as a sanity check."
> > > > >     - After: "DL research efforts aimed at surpassing GBDT should include enough datasets where GBDT outperforms DL solutions in order to avoid the bias towards DL-friendly problems"
> > > > > - L235
> > > > >     - Before: "FT-Transformer is superior to ResNet on heterogeneous data and performs on par with ResNet on non-heterogeneous data (see section 3.3). We develop this intuition further in section 4.1."
> > > > >     - After: "FT-Transformer outperforms ResNet on the problems where GBDT is superior to ResNet and performs on par with ResNet on the rest of the tasks. We develop this intuition further in section 4.1."
> > > > > - L243
> > > > >     - Before: "on heterogeneous data, even the default configuration of FT-Transformer already performs well and beats most competitors"
> > > > >     - After: "on the tasks where GBDT is the best solution, even the default configuration of FT-Transformer already performs well and beats most competitors"
> > > > > - L253
> > > > >     - Before: "Once hyperparameters are properly tuned, GBDTs start dominating on the heterogeneous datasets, while FT-Transformer keeps leadership on DNN-friendly problems"
> > > > >     - After: "Once hyperparameters are properly tuned, GBDTs start dominating on a significant portion of tasks, while FT-Transformer keeps leadership on the rest of the problems."
> > > > >
> > > > >
> > > > > ### Conclusion
> > > > > We hope that the proposed changes address the Reviewer's concerns. We will be glad to hear the feedback.

---

> > > > > > ### Comment · Reviewer_rxXz · 2021-08-27
> > > > > > **Response to Reviewer rxXz**
> > > > > >
> > > > > > Dear authors,
> > > > > >
> > > > > > thanks for your swift reply.
> > > > > > I like all proposed changes and will further increase my score to a 7.
> > > > > > I hope that you follow through with all the changes you propose here, and I would especially suggest that you take Points 1-4 of the rebuttal and add them to the paper.
> > > > > >
> > > > > > I don't quite know what the right way to bring this up is, but I think that the writing in the paper could be improved from a grammatical/legibility perspective. (For example the first sentence of the abstract.) I think it would be in your best interest to get a second opinion/proof-reading of before a camera-ready version of the paper.
> > > > > >
> > > > > > All the best!

---

> ### Author Response · Authors · 2021-08-10
> **Response to Reviewer rxxZ (Part 2)**
>
> ### 2. Experiments
>
> > For the Adult dataset, I have compared the reported performances in this paper to publically available results at ... which shows a large 0.1 difference between the top reported AUROC values
>
> Thanks, we identify an issue in our pipeline that affected the results on Adult and we provide the updated results in the table below (We report mean accuracy and standard deviations over 15 random seeds). The numbers for all other datasets are correct.
>
> | SNN            |     TabNet     | GrowNet        |      DCN2      |    AutoInt     |      MLP       |      NODE      |     ResNet     | FT-Transformer |    CatBoost    |    XGBoost     |
> |:-------------- |:--------------:|:-------------- |:--------------:|:--------------:|:--------------:|:--------------:|:--------------:|:--------------:|:--------------:|:--------------:|
> | 0.857 (0.0012) | 0.846 (0.0068) | 0.857 (0.0012) | 0.858 (0.0009) | 0.858 (0.0011) | 0.851 (0.0016) | 0.851 (0.0020) | 0.854 (0.0016) | 0.859 (0.0016) | 0.873 (0.0012) | 0.872 (0.0005) |
>
> > Why not performance in terms of average rank order across models directly, instead of eyeballing things?
>
> We provide the ranks below, however, *we believe that average rank can be a misleading metric for Tabular Data problems*. In fact, researchers come to similar conclusions for other fields as well (`[7]`, see section 3 and, in particular, section 3.2). We think that one of the key properties of Tabular Data problems is that every problem is, in some sense, *unique*. Ranking completely removes this property and makes it impossible to:
> - see if any models demonstrate unusual behavior on certain types of tasks and/or input features
> - (for readers and especially practitioners) understand if the ranking results are relevant to actual downstream tasks (or maybe profit is negligible)
> - see the scale of performance gaps on different tasks; since we aim to identify good baselines, we cannot recommend models that demonstrate disappointing performance on certain tasks
>
> Additionally, avoiding average ranks allowed us to notice the interesting story in tables 1 and 2 in the main text that eventually became POINT-4 (please, see the global response above).
>
> Now, we provide the ranks:
>
> | FT-Transformer |   ResNet    |  CatBoost   |  XGBoost   |    NODE     |    DCN2    |     MLP     |   AutoInt   |     SNN     |   GrowNet   |   TabNet    |
> |:-------------- |:-----------:|:-----------:|:----------:|:-----------:|:----------:|:-----------:|:-----------:|:-----------:|:-----------:|:-----------:|
> | 1.18 (0.833)   | 3.18 (2.29) | 3.18 (2.79) | 4.0 (4.02) | 4.91 (2.91) | 5.0 (2.73) | 5.09 (2.27) | 6.45 (1.72) | 6.73 (1.71) | 7.55 (2.35) | 7.73 (2.53) |
>
> The obtained ranking is a good illustration of how dangerous it is to use ranking: our "benchmark", as tables 1 and 2 in the main text show, is biased towards DL-friendly problems and the obtained ranking does not tell this crucial detail, which may lead to a false conclusion such as "DL models outperform GBDT".
>
> > Additionally, I would have preferred for the authors to also quantify variance across different splits
>
> We conducted the requested experiment for the *smallest* of our datasets: California Housing (20K objects, 8 features):
> - we prepared five disjoin folds and used each fold as the test set exactly once
> - we applied the tuning protocol to four algorithms on each of the five splits
> - we report the mean RMSE (with std) over the five obtained models
>
> |                                     | XGBoost       | MLP           | ResNet        | FT-Transformer |
> |:----------------------------------- |:------------- |:------------- |:------------- |:-------------- |
> | 5 folds                             | 0.445 (0.014) | 0.499 (0.008) | 0.489 (0.008) | 0.467 (0.012)  |
> | 1 split (values from the main text) | 0.443 (0.002) | 0.494 (0.004) | 0.487 (0.005) | 0.464 (0.005)  |
>
> Quick summary:
> - the standard deviations are larger than for the one split, which is expected
> - however, the mean values are very similar to those for the one split
> - given that all other datasets are even larger, we believe that the results reported in the main text are reliable
>
> ### 3. FT-Transformer
>
> > I feel that making a big deal of the FT-Transformer as a contribution generally hurts this paper. ... Adding biases to the encoding sounds like a one line code change
>
> We feel that our message was misinterpreted. Please, see POINT-2, POINT-3 and POINT-4 in the global response above.
>
> > The differences to AutoInt are small, despite what the authors claim ...
>
> We do not agree with such an interpretation. Please, note that:
> - we do not claim to be the first to utilize self-attention (L215, L291)
> - this is why we include AutoInt in the **Ablation Study**
> - we say that the design of AutoInt is different from FT-Transformer **in its details** (L295)
> - we empirically demonstrate that this difference matters
>
> > 0.01 improvement
>
> It is unclear to us, how this "0.01" number was obtained. Here are the differences inferred from Table 3 (re ~ regression, cl ~ classification):
>
> | CA (re) | HE (cl) | JA (cl) | HI (cl) | AL (cl) | YE (re) | CO (cl) | MI (re) |
> |:-------:|:-------:|:-------:|:-------:|:-------:|:-------:|:-------:|:------- |
> |  0.015  |  0.019  |  0.015  |  0.002  |  0.015  |  0.055  |  0.033  | 0.005   |
>
> Moreover, the meaning of "0.01" strongly depends on a context, which is usually defined on a per-dataset basis by gaps distributions and standard deviations of metrics:
> - "California Housing (CA)": 0.01 is just a significant improvement
> - "Microsoft (MI)": 0.01 is a big gap (0.005 in the table above is a good improvement)
> - "Year (YE)": 0.01 is a negligible improvement
> - and 0.01 is a good improvement for all classification tasks
>
> > Adding the mask tokens at input is more interesting but is not ablated
>
> There are no "mask tokens" in FT-Transformer and we ablate the only part of the architecture that is designed by us. Please, see the detailed description of the architecture in section 3.6 and the Ablation Study (section 4.2). Quick summary:
> - the backbone is just the "PreNorm" variant `[8]` of the original Transformer `[9]`
> - the inference mechanism is taken from BERT `[10]`
> - the Feature Tokenizer is proposed by us and, in our opinion, it has just one part that deserves ablation, which *is* ablated in section 4.2
>
> To sum up, we ablate things that we introduce and we do NOT ablate things that were inherited from well-known battle-tested architectures.

---

> ### Author Response · Authors · 2021-08-10
> **Response to Reviewer rxxZ (Part 3)**
>
> ### 4. Architecture Bias Experiments
>
> Before answering specific questions, we feel that a quick summary of section 4.1 can be helpful:
> 1. L235-L237 describe the motivation for additional analysis on the difference between the two strong DL models (ResNet and FT-Transformer)
> 2. in section 4.1, we design a sequence of tasks where the first one is ResNet-friendly, the last one is GBDT-friendly and all intermidiate ones are the result of gradual interpolation between the first task and the last task. Note that the CatBoost and ResNet lines in Figure 3 in the main text demonstrate that we indeed constructed the desired sequence of tasks.
> 3. now, the main hypothesis: if the observation made in L235-L237 is not a random coincidence, then the **relative performance** of FT-Transformer with respect to ResNet should improve as we gradually move from ResNet-friendly tasks to GBDT-friendly tasks.
>
> > What's benefit of linearly interpolating targets?
>
> The purpose of interpolation is not "benefit" of any kind, but a gradual move from more DL-friendly tasks to more GBDT-friendly tasks.
>
> > Why not generate data from the transformer as well?
>
> It turns out that the phenomenon that we are exploring in section 4.1 can be illustrated just with MLP, so we kept things simple.
>
> > Why is XGBoost not included here?
>
> Tables 1 and 2 in the main text do not reveal any significant difference between CatBoost and XGBoost so, for our purposes, picking just one of them is enough. Moreover, those tables show that the default CatBoost is usually a better choice, so this detail defined our final choice.
>
> > Why not extend this to all models in the spirit of the 'big comparison' paper?
>
> Please, see the summary above. We did not aim to analyze *all* inductive biases, only those that demonstrated the strongest empirical performance. At the same time, the experiment can be extended to any number of task sequences and models. However, doing this would require much compute and is out of scope for our work (since the goal is to identify and analyze the *strongest* baselines).
>
> > Fig. 3 seems to suggest that CatBoost can accommodate all data while ResNets can't acccomodate CatBoost-data. How does this relate to the experiments
>
> First, in the context of section 4.1 absolute performance is less important than the relative performance (please, see the summary above). Second, constructing GBDT-friendly task turned out to be much easier: an ensemble of random shallow trees perfectly fits the inductive bias of GBDT. We failed to design such a "perfect" task for ResNet, however, the chosen "imperfect" task also did its job just fine.
>
> > Can this be related to the statements about hetero-/homogeneous datasets?
>
> Yes, this is the main purpose of section 4.1 and we make this observation in L286-L288.
>
> ### Limitations And Societal Impact
>
> > It is unclear to me how this relates to societal impact.
>
> It has a *direct* relation to societal impact. If tomorrow every ML pipeline working with Tabular Data is switched to FT-Transformer, we will see a tremendous relative increase in CO2 emissions generated by those pipelines. Note that we are aware of other aspects of societal impact (such as fairness, data memoization etc.), but all of those are *default* for Deep Learning models today (i.e. something we should *always* be thinking about). This is why we focused on the problem that is more *specific* to the models that are discussed in our work.
>
> References:
> - `[1]`: G. Klambauer, T. Unterthiner, A. Mayr, and S. Hochreiter. Self-normalizing neural networks. In NIPS, 2017.
> - `[2]`: Baldi, P., P. Sadowski, and D. Whiteson. Searching for Exotic Particles in High-energy Physics with Deep Learning. Nature Communications 5 (July 2, 2014)
> - `[3]`: Sergei Popov, Stanislav Morozov, Artem Babenko: Neural Oblivious Decision Ensembles for Deep Learning on Tabular Data. ICLR 2020
> - `[4]`: Sarkhan Badirli, Xuanqing Liu, Zhengming Xing, Avradeep Bhowmik, Sathiya S. Keerthi: Gradient Boosting Neural Networks: GrowNet. CoRR abs/2002.07971 (2020)
> - `[5]`: Sercan Ö. Arik, Tomas Pfister: TabNet: Attentive Interpretable Tabular Learning. AAAI 2021: 6679-6687
> - `[6]`: Jannik Kossen, Neil Band, Clare Lyle, Aidan N. Gomez, Tom Rainforth, Yarin Gal: Self-Attention Between Datapoints: Going Beyond Individual Input-Output Pairs in Deep Learning. CoRR abs/2106.02584 (2021)
> - `[7]`: Mostafa Dehghani, Yi Tay, Alexey A. Gritsenko, Zhe Zhao, Neil Houlsby, Fernando Diaz, Donald Metzler, Oriol Vinyals: The Benchmark Lottery. CoRR abs/2107.07002 (2021)
> - `[8]`: Qiang Wang, Bei Li, Tong Xiao, Jingbo Zhu, Changliang Li, Derek F. Wong, Lidia S. Chao: Learning Deep Transformer Models for Machine Translation. ACL (1) 2019: 1810-1822
> - `[9]`: Ashish Vaswani, Noam Shazeer, Niki Parmar, Jakob Uszkoreit, Llion Jones, Aidan N. Gomez, Lukasz Kaiser, Illia Polosukhin: Attention is All you Need. NIPS 2017: 5998-6008
> - `[10]`: Jacob Devlin, Ming-Wei Chang, Kenton Lee, Kristina Toutanova: BERT: Pre-training of Deep Bidirectional Transformers for Language Understanding. NAACL-HLT (1) 2019: 4171-4186

---

### Official Review · Reviewer_KYAo · 2021-07-15

**Rating:** 7
**Confidence:** 4

**Summary:**

- The authors conduct extensive experiments on tabular data learning with various datasets and baselines.
- The authors discover some important takeaways for tabular learning such as when we should use shallow or deep models.
- The authors propose a novel tabular learning framework which shows the state-of-the-art performance in various datasets.

**Ethical Concerns:**

There is no ethical concern.

**Limitations And Societal Impact:**

The authors properly describe the limitations of the paper and baselines.
However, it would be better if the authors can somehow tone down for the discovery and conclusions because those are only supported by the empirical results.

**Main Review:**

1. Impactful paper
- This paper tackles an important problem which has significant attention from both academia and industry.
- Constructing standard experimental settings, datasets, metrics, and baselines in this field would be highly beneficial for future research in this area.
- Also, it is important for practitioners which model works well for which condition. When we should use which model is a critical question for many practitioners.
- The published codebase is clear and easy to understand and reproduce.

2. ResNet
- Have any previous works tried ResNet type of architecture for tabular data?
- If yes, it would be good to cite those papers.
- If not, this would be another contribution of this paper.

3. Experimental details
- How to treat the categorical variables? One-hot encoding?
- Would be good to describe the number of feature ranges in the main manuscript.
- Can we use "patience" as another hyper-parameter? How 16 is determined?

4. Sanity check
- It would be good if the authors can provide whether the implementation of each model can reproduce the results in each paper.
- For instance, the authors can utilize at least one data per each baseline and check whether the results can be reproduced.
- This would be important because it can be a sanity check whether the detailed implementation is accurate for those baselines.

5. Etc
- It would be good if the authors can clarify which dataset is regression, binary, and multi-class classification tasks in results tables.
- I would like to know whether the performance of an FT-transformer is dependent on the input order of features.

**Time Spent Reviewing:**

3 hours

---

> ### Author Response · Authors · 2021-08-10
> **Response to Reviewer KYAo**
>
> Thank you for the review!
>
> > Have any previous works tried ResNet type of architecture for tabular data? If yes, it would be good to cite those papers
>
> Yes, and we cite all the papers using ResNet-like models for Tabular Data that we are aware of:
> - `[1]`. However, we could not find the exact implementation. Moreover, our ResNet demonstrates better relative performance compared to the SNN architecture from `[1]` and we share our implementation so it can serve as a good baseline
>
> We are also aware of `[2]`. The field is different (Natural Language Processing), but the architecture is relevant. We plan to cite this paper as well as an additional motivation for including ResNet in the comparison.
>
> > How to treat the categorical variables? One-hot encoding?
>
> For every algorithm, we follow the best available strategy:
> - XGBoost: One-Hot Encoding (OHE)
> - CatBoost: it has a built-in support of categorical features, so we pass them as is
> - All DL models, except for the FT-Transformer: we use embeddings (which in its formal description imply the usage of OHE)
> - FT-Transformer: the method of processing categorical features is described in detail in section 3.6 of the main text (the "Feature Tokenizer" paragraph); see also the illustration in Figure 2a in the main text.
>
> > Can we use "patience" as another hyper-parameter? How 16 is determined?
>
> While the theoretical answer to the first question is "yes", we instead try to minimize the influence of early stopping as much as possible so that all models reach most of their potential without the risk of being stopped too early. We achieve that by setting the patience to 16, which is a pretty high value (the value "16" is not special and was picked as a power of 2). Moreover, we hypothesize that including the patience in the hyperparameter optimization will require a much higher budget in terms of optimization trials, since the trials with non-optimal patience will lead to low performance regardless of how good other hyperparameters are, which is itself detrimental to the tuning in general.
>
> > I would like to know whether the performance of an FT-transformer is dependent on the input order of features
>
> - "Yes" for the "Feature Tokenizer" module, since there are separate trainable parameters for each input feature
> - "No" for the "Transformer" module: once all input features are tokenized, the tokens can be passed to the "Transformer" module in any order
>
> Please, see section 3.6 in the main text for the full description of FT-Transformer.
>
> References:
> - `[1]`: G. Klambauer, T. Unterthiner, A. Mayr, and S. Hochreiter. Self-normalizing neural networks. In NIPS, 2017.
> - `[2]`: Simeng Sun, Mohit Iyyer: Revisiting Simple Neural Probabilistic Language Models. NAACL-HLT 2021: 5181-5188

---

### Official Review · Reviewer_RbMa · 2021-07-16

**Rating:** 7
**Confidence:** 5

**Summary:**

This paper set up a standard benchmark for recent works on NN for tabular data, and GBDT, to check whether these works indeed outperform GBDT or not. Besides, two new models, ResNet-like MLP and transformer-based models are proposed, and seem can outperform previous works.

**Limitations And Societal Impact:**

yes

**Main Review:**

Strengths:

- The standard benchmark is very helpful. With it, we can know which models are better. I believe this can boost the research for the tabular data learning domain.

- The design of the FT-Transformer is simple and solid. Pre-ln is a good choice. The bias term in Tokenizer seems helpful.

Weakness & questions:
- As both xgboost and catboost are set to level-wise trees, it would be better to have the results of the leaf-wise tree, like LightGBM.

- The selected benchmark datasets are quite messy. It will be better to categorize them by applications/scenarios. Besides, as tabular data is widely used in ads click prediction and recommenders systems, it would be better if the benchmark can cover these two types of tasks.

- For the experiment results, it seems GBDT is not outperformed by NN in tabular data. I notice there are "heterogeneous" and "non-heterogeneous" datasets in this paper. Obviously, the heterogeneous dataset is more like the tabular data in the real-world (Although multi-modal data is more common, I think we should focus on heterogeneous tabular data, as NN is already very good in other data. As for high-dimensional multi-class data, it actually is not common in real-world applications with tabular data.). From the results, GBDT is still the best one in heterogeneous data. To avoid confusion, I think the author should make the table clear (even use two tables), let readers know there are two kinds of datasets.

- Based on 3, I think the author should make the claim more carefully.  In my opinion, the heterogeneous dataset is more like tabular data. So the sentence like "we show that the choice between GBDT and DL models highly depends on data and there is still no universally superior solution." looks ambiguous.

- Do you have a plan to host a leaderboard, like GLUE in NLP?

**Time Spent Reviewing:**

7

---

> ### Author Response · Authors · 2021-08-10
> **Response to Reviewer RbMa**
>
> Thank you for your thoughtful review!
>
> > As both xgboost and catboost are set to level-wise trees, it would be better to have the results of the leaf-wise tree, like LightGBM.
>
> We provide the results for LightGBM on the three datasets below. Overall, the results reported in the paper show that DL models are not yet at the point where they can compete with GBDT on GBDT-friendly datasets, so we refrained from adding more GBDT solutions. Nevertheless, if/when DL models start beating *some* GBDT solutions on GBDT-friendly datasets, we will definitely need to include *all* GBDT solutions to ensure that the family of GBDT algorithms is fully represented.
>
> CA = California Housing, AD = Adult, HI = Higgs
>
> |     | LightGBM       | CatBoost       | XGBoost        | MLP            | ResNet         | FT-Transformer |
> |:--- |:-------------- |:-------------- |:-------------- |:-------------- |:-------------- |:-------------- |
> | CA  | 0.432 (0.0035) | 0.431 (0.0016) | 0.433 (0.0016) | 0.494 (0.0036) | 0.487 (0.0053) | 0.464 (0.0051) |
> | AD  | 0.872 (0.0003) | 0.873 (0.0012) | 0.872 (0.0005) | 0.851 (0.0016) | 0.854 (0.0016) | 0.859 (0.0016) |
> | HI  | 0.724 (0.0014) | 0.725 (0.0016) | 0.727 (0.0011) | 0.722 (0.0024) | 0.726 (0.0018) | 0.728 (0.0019) |
>
> The table demonstrates that LightGBM performs similarly to other GBDT. Three datasets are not a lot, but the results are in line with our expectations.
>
> > as tabular data is widely used in ads click prediction and recommenders systems, it would be better if the benchmark can cover these two types of tasks.
>
> The following table demonstrates results for the KDD challenge on ads click prediction. Admittedly, the models do not differ much from each other, so we should look for better benchmarks for future work.
>
> |                          |                 |
> |:------------------------ |:--------------- |
> | catboost                 | 0.6635 (0.0002) |
> | xgboost                  | 0.6399 (0.0006) |
> | mlp                      | 0.6617 (0.0006) |
> | resnet                   | 0.6612 (0.0007) |
> | ft_transformer (default) | 0.6610 (0.0007) |
> | ft_transformer           | 0.6606 (0.0009) |
> | snn                      | 0.6613 (0.0006) |
> | autoint                  | 0.6614 (0.0005) |
> | grownet                  | 0.6614 (0.0009) |
> | dcn2                     | 0.6615 (0.0003) |
>
> > In my opinion, the heterogeneous dataset is more like tabular data.
>
> Personally, we fully share this intuition and believe that "heterogeneous" data should be prioritized for Tabular DL research. However, we tried to avoid our personal bias and included "non-heterogeneous" datasets as well. As a side note: we think of replacing the term "heterogeneous" with "GBDT-friendly", because it is hard to give a formal definition to the term "heterogeneous".
>
> > Do you have a plan to host a leaderboard, like GLUE in NLP?
>
> As of now:
> - we plan to share the dataset splits, baselines, the tuning protocol and the evaluation protocol of high usability
> - we plan to continue adding more good (especially GBDT-friendly) datasets as we find them
> - we have not yet made a decision about a high-quality automated leaderboard (which is a big commitment)

---

### Official Review · Reviewer_hjQ3 · 2021-07-16

**Rating:** 7
**Confidence:** 4

**Summary:**

This paper investigates the performance of deep learning models on tabular data and compares it with traditional models such as gradient boosted decision trees. It shows that ResNet is good baseline for deep learning models on tabular data. It also proposes a new attention based architecture which surpasses all the deep learning methods mentioned in the paper.

**Limitations And Societal Impact:**

Yes they have.

**Main Review:**

Pros:
- The study is quite comprehensive covering a range of datasets
- The paper analyzes multiple models from each category
- The proposed transformer based model surpasses other models in most tasks and is good baseline for future tasks
- All models are explained in detail in the supplementary materials for reproducibility
- Training runtimes are also compared

Cons:
- FT-Transformer, the proposed method, is more complex and takes longer to train on average than ResNet

Overall, the paper is very useful and provides a great overview of the different approaches for tabular data. By proposing a new method which surpasses the approaches mentioned in the paper, it even adds to the existing body of baselines for future work in the area. The code is open sourced and the models are discussed in great detail in the Appendix.


**Time Spent Reviewing:**

4

---

> ### Author Response · Authors · 2021-08-10
> **Response to Reviewer hjQ3**
>
> Thank you for the review and the positive feedback! We agree that the FT-Transformer is more complex and requires more compute. We believe that making it more efficient is a promising research direction.

---

### Official Review · Reviewer_wxKJ · 2021-07-20

**Rating:** 7
**Confidence:** 4

**Summary:**

This paper
1) conducts a careful comparison of many deep learning methods for tabular learning
2) introduces a new baseline method for tabular learning (ResNet)
3) introduces a new fancy architecture for tabular learning (FT-Transformer)

**Limitations And Societal Impact:**

Fine.

**Main Review:**

The paper is very closely related to (and partially superceded by) [1],
but since [1] appeared on arXiv after the paper was submitted
I will try to evaluate the current paper as though [1] did not exist.

The writing is clear and the experiments seem careful and thoughtful.
Contributions 1) and 2) are important and useful.
Unfortunately, the new baseline 2) rather steals the thunder from the
new architecture 3).
The experiments don't make it clear that the new architecture is worthwhile.
The authors are very straightforward about the limitations of FT-transformer
(and its much greater cost compared to ResNet), which I appreciate.

Hyperparameter tuning is a key issue for a comparison.
Given more time [or, more generally, computational resources]
and more parameters (and enough data), DL models tend to improve.
Were some of these methods given more time than others?
(Including time to choose hyperparameters?)
I'd like to have seen performance as a function of time, or after a fixed amount of time.

As a more narrow question, why Optuna? Does it matter?
(Looking into the future, [1] uses BOHB [2]. Do you think swapping the hyperparameter tuner would affect your results?)

* 185 "in the ensembling mode" -> define
* 202 ALOI -> meaning of tilde is confusing, say "The ALOI dataset has nearly 1000 classes".
* I found it easy to miss the definition of "heterogeneous". Maybe use a definition environment for it?
* 240: FT-transformer is not sensitive to hyperparameters - I'd like to see more experiments demonstrating this, and looking at the quality of model as a function of training time (including hyperparameter optimization time)

[1] Kadra, Arlind, et al. "Regularization is all you Need: Simple Neural Nets can Excel on Tabular Data." arXiv preprint arXiv:2106.11189 (2021).
[2] Falkner, Stefan, Aaron Klein, and Frank Hutter. "BOHB: Robust and efficient hyperparameter optimization at scale." International Conference on Machine Learning. PMLR, 2018.


**Time Spent Reviewing:**

1

---

> ### Author Response · Authors · 2021-08-10
> **Response to Reviewer wxKJ**
>
> Thank you for the detailed review!
>
> > Were some of these methods given more time than others? (Including time to choose hyperparameters?)
>
> We did not impose any *time* budget restrictions on any algorithms. Instead, we allocated a pretty generous budget in terms of *trials for hyperparameters optimization* (100 for most cases, 50 for slower algorithms) for all algorithms to make sure that all of them reach most of their potential. The same applies to *evaluation*, where each algorithm was trained with high early stopping patience to lower the risk of finishing training too early.
>
> > I'd like to have seen performance as a function of time, or after a fixed amount of time.
>
> We conducted the following experiment:
> - we consider four algorithms: (1) XGBoost as a fast GBDT implementation (2) MLP as the fastest and simplest DL model (3) ResNet as a stronger but slower DL model (4) FT-Transformer as the strongest and the slowest DL model
> - we consider three datasets (see below)
> - on each dataset, for each algorithm, we run *five* independent (five random seeds) hyperparameter optimizations; each optimization consists of 200 iterations;
> - for every time budget:
>   - we pick the best model identified by Optuna on the **validation** set if spent no more than this time budget
>   - we report the following metrics for this model:
>     - the mean score on the **test** set (we do not report std due to the lack of space, but those are similar to the ones from the main texts)
>     - the mean number of Optuna iterations performed during the time budget (note, this is NOT the Optuna iteration of the chosen model)
> - we use "!" to mark the best DL-solution and "!!" to mark the overall best solution (it turns out that this does NOT depend on the budget)
>
> California Housing (RMSE, 20640 objects, 8 features):
>
> |                  | 0.25h      | 0.5h       | 1h          | 2h          | 3h          | 4h          | 5h          | 6h          |
> |:---------------- |:---------- |:---------- |:----------- |:----------- |:----------- |:----------- |:----------- |:----------- |
> | XGBoost !!       | 0.437 / 31 | 0.436 / 56 | 0.434 / 120 | 0.433 / 252 | 0.433 / 410 | 0.432 / 557 | 0.433 / 719 | 0.432 / 867 |
> | MLP              | 0.503 / 16 | 0.496 / 42 | 0.493 / 103 | 0.488 / 230 | 0.489 / 349 | 0.489 / 466 | 0.488 / 596 | 0.488 / 724 |
> | ResNet           | 0.488 / 7  | 0.487 / 15 | 0.483 / 30  | 0.481 / 64  | 0.482 / 101 | 0.482 / 131 | 0.482 / 164 | 0.484 / 197 |
> | FT-Transformer ! | 0.466 / 4  | 0.464 / 9  | 0.465 / 20  | 0.460 / 47  | 0.458 / 74  | 0.458 / 99  | 0.457 / 124 | 0.459 / 153 |
>
> Adult (accuracy, 48842 objects, 14 features):
>
> |                  | 0.25h       | 0.5h        | 1h          | 2h           | 3h           | 4h           | 5h           | 6h           |
> |:---------------- |:----------- |:----------- |:----------- |:------------ |:------------ |:------------ |:------------ |:------------ |
> | XGBoost !!       | 0.871 / 165 | 0.873 / 311 | 0.872 / 638 | 0.872 / 1296 | 0.872 / 1927 | 0.872 / 2478 | 0.872 / 2999 | 0.872 / 3500 |
> | MLP              | 0.856 / 20  | 0.857 / 37  | 0.858 / 71  | 0.857 / 130  | 0.856 / 190  | 0.856 / 247  | 0.856 / 310  | 0.856 / 375  |
> | ResNet           | 0.856 / 8   | 0.854 / 16  | 0.854 / 32  | 0.856 / 69   | 0.855 / 105  | 0.855 / 140  | 0.856 / 174  | 0.855 / 208  |
> | FT-Transformer ! | 0.861 / 6   | 0.860 / 12  | 0.859 / 27  | 0.859 / 52   | 0.860 / 78   | 0.860 / 99   | 0.860 / 125  | 0.860 / 148  |
>
>
> Higgs (accuracy, 98050 objects, 28 features):
>
> |                   | 0.25h      | 0.5h        | 1h          | 2h          | 3h          | 4h           | 5h           | 6h           |
> |:----------------- |:---------- |:----------- |:----------- |:----------- |:----------- |:------------ |:------------ |:------------ |
> | XGBoost           | 0.725 / 88 | 0.725 / 153 | 0.724 / 291 | 0.725 / 573 | 0.725 / 823 | 0.726 / 1069 | 0.725 / 1318 | 0.725 / 1559 |
> | MLP               | 0.721 / 16 | 0.720 / 29  | 0.723 / 62  | 0.722 / 137 | 0.724 / 220 | 0.723 / 300  | 0.724 / 375  | 0.724 / 447  |
> | ResNet            | 0.724 / 8  | 0.727 / 14  | 0.727 / 32  | 0.728 / 61  | 0.728 / 84  | 0.728 / 107  | 0.728 / 132  | 0.728 / 154  |
> | FT-Transformer !! | 0.727 / 2  | 0.729 / 5   | 0.728 / 12  | 0.728 / 23  | 0.729 / 34  | 0.729 / 44   | 0.730 / 56   | 0.729 / 66   |
>
> Quick summary:
> 1. interestingly, FT-Transformer achieves good metrics just after several *randomly* sampled configurations (Optuna performs simple random sampling during the first 10 (default) iterations)
> 2. FT-Transformer is slower to train, which is expected
> 3. extended tuning (in terms of *iterations*) for other algorithms does not lead to any meaningful improvements
>
> > As a more narrow question, why Optuna? Does it matter? ... Do you think swapping the hyperparameter tuner would affect your results?
>
> We simply took the most cited paper `[1]` which is additionally supported by the most popular GitHub repository related to hyperparameters optimization algorithms. The recent publication `[2]` demonstrates the superiority of Bayesian optimization (BO) to random search (RS), however, the difference between variants of BO is much less significant than between BO and RS (Table 2, page 7 in `[2]`). So we believe that we made a reasonable "default" choice and more sophisticated algorithms are unlikely to affect the conclusions. Thank you for the question, we are glad that our effort to properly tune the algorithms gets attention. We get additional motivation to keep an eye on the latest algorithms and try new ones.
>
> > 240 FT-transformer is not sensitive to hyperparameters - I'd like to see more experiments demonstrating this
>
> We thank the reviewer for paying attention to this point.
> - we meant that there is one specific dataset-agnostic configuration (which we call "default" in the main text) of FT-Transformer that yields competitive performance out-of-the-box
> - the experiment with time budgets (see above) makes us think that FT-Transformer may be good in terms of sensitivity to hyperparameters, but we do not provide more rigorous evidence.
>
> Will clarify this point in the revision.
>
> > 185 "in the ensembling mode" -> define ... 202 ALOI -> meaning of tilde is confusing, say "The ALOI dataset has nearly 1000 classes".
>
> Thank you for pointing to this, we will fix these issues in the next revision.
>
> References:
> - `[1]` Takuya Akiba, Shotaro Sano, Toshihiko Yanase, Takeru Ohta, Masanori Koyama: Optuna: A Next-generation Hyperparameter Optimization Framework. KDD 2019: 2623-2631
> - `[2]` Ryan Turner, David Eriksson, Michael McCourt, Juha Kiili, Eero Laaksonen, Zhen Xu, Isabelle Guyon: Bayesian Optimization is Superior to Random Search for Machine Learning Hyperparameter Tuning: Analysis of the Black-Box Optimization Challenge 2020. CoRR abs/2104.10201 (2021)

---

### Author Response · Authors · 2021-08-10
**Global response to the reviewers**

(`R1` = `Reviewer wxKJ`, `R2` = `Reviewer hjQ3`, `R3` = `Reviewer RbMa`, `R4` = `Reviewer KYAo`, `R5` = `Reviewer rxXz`)

We thank all the reviewers for the feedback! We are glad that they found our comparison to be an important contribution (`R1`, `R2`, `R3`, `R4`) and found it to be solid (`R1`, `R2`, `R3`, `R4`). We are also encouraged that the reviewers found the new baselines to be useful/reasonable (ResNet: `R1`; FT-Transformer: `R2`, `R3`; baselines in general: `R4`). We are also glad that our effort to make the experiments reproducible is appreciated (`R2`, `R4`).

We have incorporated the feedback and answered the questions in the individual responses. Additionally, we would like to provide some common points that clarify our motivation, goals and contributions.

- **POINT-1**: as of now, in the field of Tabular DL, it is not clear what architectures can be considered as reasonable baselines. As a result, the two most often used baselines are MLP and TabNet `[1,2,3,4,5,6,7,8]`. Unfortunately, as it turns out, they both do not represent a significant challenge.
- **POINT-2**: we aim to raise the bar of DL baselines and improve the situation described in **POINT-1**. We aim to identify a minimal set of such DL baselines for Tabular Data problems, that are: (1) simple (2) demonstrate robust performance across many tasks (3) set a high bar for future work on Tabular DL.
- **POINT-3**: we do NOT aim to propose *conceptually novel* architectures. In our opinion, the tabular DL is currently overwhelmed with different architectures but lacks simple and effective baselines. In this regard, we aim to: (1) design such baselines carefully (2) make implementations public (3) explicitly describe distinctive properties of the proposed baselines (including training costs).
- **POINT-4**: we discover a property of FT-Transformer that *we believe can become an important milestone* on the path to achieving the GBDT-level performance on those problems where GBDT currently dominates. Namely, *FT-Transformer tends to outperform other DL architectures exactly on GBDT-friendly datasets* (i.e. on those datasets where GBDT is superior to DL solutions). Please, see section 4.1 in the main text for additional analysis on this phenomenon.

References:
- `[1]` Sarkhan Badirli, Xuanqing Liu, Zhengming Xing, Avradeep Bhowmik, Sathiya S. Keerthi: Gradient Boosting Neural Networks: GrowNet. CoRR abs/2002.07971 (2020)
- `[2]` Liran Katzir, Gal Elidan, Ran El-Yaniv: Net-DNF: Effective Deep Modeling of Tabular Data. ICLR 2021
- `[3]` Sergei Popov, Stanislav Morozov, Artem Babenko: Neural Oblivious Decision Ensembles for Deep Learning on Tabular Data. ICLR 2020
- `[4]` Arlind Kadra, Marius Lindauer, Frank Hutter, Josif Grabocka: Regularization is all you Need: Simple Neural Nets can Excel on Tabular Data. CoRR abs/2106.11189 (2021)
- `[5]` Jannik Kossen, Neil Band, Clare Lyle, Aidan N. Gomez, Tom Rainforth, Yarin Gal: Self-Attention Between Datapoints: Going Beyond Individual Input-Output Pairs in Deep Learning.
- `[6]` Gowthami Somepalli, Micah Goldblum, Avi Schwarzschild, C. Bayan Bruss, Tom Goldstein: SAINT: Improved Neural Networks for Tabular Data via Row Attention and Contrastive Pre-Training. CoRR abs/2106.01342 (2021)
- `[7]` Xin Huang, Ashish Khetan, Milan Cvitkovic, Zohar S. Karnin: TabTransformer: Tabular Data Modeling Using Contextual Embeddings. CoRR abs/2012.06678 (2020)
- `[8]` Ravid Shwartz-Ziv, Amitai Armon: Tabular Data: Deep Learning is Not All You Need. CoRR abs/2106.03253 (2021)

---

### Decision · Program_Chairs · 2021-09-27

**Decision:**

Accept (Poster)

**Comment:**

All reviewers have agreed on the positive aspects of the paper, and its important contributions for tabular deep learning. There were significant additions by the authors during the review process, particularly extra experimental results, method motivations, clarification of experimental setups and hyperparameter tuning. It is important to reflect these in the final version of the paper. Finally, as discussed, it would be an impactful addition to open-source the benchmarking framework.